# Surprising combinations of research contents and contexts are related to impact and emerge with scientific outsiders from distant disciplines

Feng Shi [1,2] & James Evans [2,3,4] ✉

We investigate the degree to which impact in science and technology is associated with surprising breakthroughs, and how those breakthroughs arise. Identifying breakthroughs across science and technology requires models that distinguish surprising from expected advances at scale. Drawing on tens of millions of research papers and patents across the life sciences, physical sciences and patented inventions, and using a hypergraph model that predicts realized combinations of research contents (article keywords) and contexts (cited journals), here we show that surprise in terms of unexpected combinations of contents and contexts predicts outsized impact (within the top 10% of citations). These surprising advances emerge across, rather than within researchers or teams—most commonly when scientists from one field publish problem-solving results to an audience from a distant field. Our approach characterizes the frontier of science and technology as a complex hypergraph drawn from high-dimensional embeddings of research contents and contexts, and offers a measure of path-breaking surprise in science and technology.

19th Century philosopher and scientist Charles Sanders Peirce argued that neither the logics of deduction nor induction alone could characterize the reasoning behind path-breaking new hypotheses in science, but rather their collision in a process he termed abduction. Abduction begins as expectations born of theory, experience or tradition become disrupted by unexpected or surprising findings[1]. Surprise stimulates scientists to forge new claims that make the surprising unsurprising. Here we empirically demonstrate across the life sciences, physical sciences, and patented invention that, following Peirce, surprising discoveries and inventions are predictors of outsized impact. But it is unclear where new hypotheses come from. One account is serendipity or making the most of surprising encounters[2,3], encapsulated in Pasteur's oft-quoted maxim "chance favors only the prepared mind"[4], but this poses a paradox. The successful scientific mind must simultaneously know enough within a scientific or technological context to be surprised at anomalies, but enough outside that context to imagine why they should not be surprising. Here we show how surprising successes systematically emerge across, rather than within researchers; most commonly when those in one field surprisingly publish problem-solving results to audiences in a distant other.

We frame surprise in science and technology as the violation of expectations held by those within a scientific field about future advance[5]. This demands that we predict the composition of future research with sufficient accuracy that what cannot be forecast will surprise the community of scientists and inventors who also compete to anticipate the future[6,7]. Our generative model deviates from previous work that simply scores a paper or patent's novelty by comparing its components to those of an average or random one[8,9]. It also deviates from work that focuses on the institutional structures that influence discovery in science[10–15]. This conceptual shift to directly

[1]TigerGraph, 3 Twin Dolphin Dr, St. 225, Redwood City, CA 94065, USA. [2]Knowledge Lab, University of Chicago, 1155 E. 60th Street #211, Chicago, IL 60637, USA. [3]Department of Sociology, University of Chicago, 1126 E. 59th St. #420, Chicago, IL 60637, USA. [4]Santa Fe Institute, 1399 Hyde Park Rd., Santa Fe, NM 87501, USA. ✉e-mail: jevans@uchicago.edu

model research outputs leads to better assessments of research novelty and its reception as surprise by researchers within scientific and technical communities. Because of this complex interplay, we use unexpected, novel, surprising, and their derivations interchangeably in this paper.

Prior literature[8,16–19] has achieved success in modeling discoveries and inventions with a combinatorial process. Our approach builds on this, drawing inspiration from recent demonstrations that beyond simple combinations, higher-order structure is critical for understanding complex networks, from transportation and neural networks[20] to foodwebs[21,22]. We model advancing science and technology with an embedding of scientific contents and contexts from biomedical and physical science articles and technology patents. Contents refer to the substance of papers and patents such as concepts and methods, while contexts refer to scientific or technological disciplines from which concepts and methods are drawn.

Distinguishing contents from contexts allows us to characterize the nature of a discovery or invention's novelty and associated surprise more precisely than before[23]. A new combination of contents may surprise because it has never succeeded before, even though it may have been considered and attempted previously in a shared context[24–27]. A new discovery or invention that cuts across divergent contexts may surprise because it has neither been attempted nor imagined before—a combination of ideas inaccessible within a single disciplinary conversation. The separate consideration of contents and contexts also allows us to contrast scientific discovery with technological search: Fields and their boundaries are clear and ever-present for scientists at all phases of scientific production, publishing and promotion, but largely invisible for technological invention and its certification in legally protected patents and marketed products.

## Results

To predict new innovations in science and technology, we develop a model that generates normal discoveries as combinations of prior knowledge[28–31]. Our model articulates a simple cognitive process: Scientists and inventors combine things together that are (1) scientifically or technologically close and (2) cognitively salient. While formally simple, this model is more effective than those from the literature and even advanced deep learning models in capturing surprise and impact (see SI for detailed comparisons). Following from this design, we model the likelihood that contents or contexts become combined in the future as a function of (1) their proximity in a latent embedding space derived from the complex network structure of prior relationship among contents and contexts and (2) their salience to scientists through prior usage frequency.

Specifically, we develop a generative hypergraph model that extends the mixed-membership stochastic block model[32] into high-dimensions, characterizing complete combinations of contents and contexts (Fig. 1). The model (1) constructs a continuous embedding for nodes from the hypergraph of contents or contexts in a given year, (2) allows that embedding to evolve stochastically, then (3) draws a new hypergraph from the updated embedding, which forms our prediction for next year's combinations in published articles[33]. This design allows us to predict which new combinations are expected to occur based on current trends in content and context. This, in turn, allows us to identify surprising combinations—those least likely according to our model—when they arise and forecast their impact (see "Methods" for details).

In this study, we apply our methodology to three major corpora of scientific knowledge and technological advance: 19,916,562 biomedical articles published between 1865 and 2009 from the MEDLINE database; 541,448 articles published between 1893 and 2013 in the physical sciences from journals published by the American Physical Society (APS), and 6,488,262 patents granted between 1979 and 2017 from the US Patent database (see "Methods" for details.) We operationalize research contents as keywords distilled within community-curated ontologies—Medical Subject Heading (MeSH) terms for MEDLINE papers, Physics and Astronomy Classification Scheme (PACS) codes for APS papers, and United States Patent Classification (USPC) codes for patents. We operationalize contexts as disciplinary journals and conferences referenced within a paper (or technology classes cited by a patent). For each dataset in each year we build a hypergraph of contents where each node corresponds to a content keyword and each hyperedge (a "link" between ≥2 nodes) to a research paper or patent that combines all such keywords. Meanwhile, for each dataset in each year we separately build a hypergraph of contexts where each node corresponds to a journal or conference (or major technological area for patents) and each hyperedge to a paper or patent that references these disciplinary contexts as sources of inspiration and influence.

Across biomedical sciences, physical sciences, and inventions, our model correctly distinguishes between a content combination that turned into a publication and a random combination more than 95% of the time for a given year when trained on data from previous years (Biomedicine: AUC = 0.98; Physics: AUC = 0.97; Inventions: AUC = 0.95). New context combinations are also predictable (Biomedicine: AUC = 0.99; Physics: AUC = 0.88; Inventions: AUC = 0.83). See "Methods" for detailed model evaluations. The successful prediction of future combinations suggests that our model inscribes a space of latent knowledge. Researchers tend to wander locally across this space in generating new papers and patents. This aligns with previous findings regarding inertia and conservative search in science[24,34–37] and gives us further confidence in the model.

With a measure of what science and technology is expected, we can assess the novelty of a combination $h$ as its improbability or surprisal[38]. Isofar as we model contents and contexts separately—a paper is simultaneously a combination of contents (e.g., curated keywords) and a combination of contexts (e.g., cited journals)—we also measure the content and context novelty of a paper separately corresponding to the suprisal of its content and context combinations, respectively.

The research our model identifies as surprising is perceived and labeled as groundbreaking by established scientists who review it for the Faculty Opinions platform—formerly Faculty of 1000 (facultyopinions.com), as detailed below (and in Supplementary Fig. 7 and SI). Surprising papers also achieve outsized impact. For example, the work of Grynkiewicz and colleagues[39] defines a novel family of chemical compounds including Fura-2, discovered to be highly fluorescent and bind to free calcium. This was a surprising discovery—pulling diverse properties together that crossed traditional field boundaries with content and context novelties in the 99th percentile and receiving more than 16,000 citations to date. Research by Yanagisawa and colleagues[40] isolated endothelin, one of the most potent vasoconstrictors at the time (1988), and employed several distinct, community spanning methods to examine its properties and mechanisms, discovering its potential as a modulator of ion channels. This discovery is in the 95th percentile of content and context novelty, receiving more than 14,000 citations since publication. By contrast, content and context novelties diverge in work by Altschul and others[41], which lies in the 97th percentile of context novelty, but only the 15th percentile of content novelty. The work used a computer system to search protein and DNA databases and did not itself represent a surprising biomedical discovery, but produced a widely used tool sourced from across the computational and bio-sciences fields, making it one of the 15 most cited articles of all time[42].

### Surprise relates to impact

We sytematically examine how surprise relates to citation impact and awards by dividing biomedical science (MEDLINE) papers into citation deciles and normalizing novelty scores, transforming them into percentiles. The decile-averaged content and context novelties increase

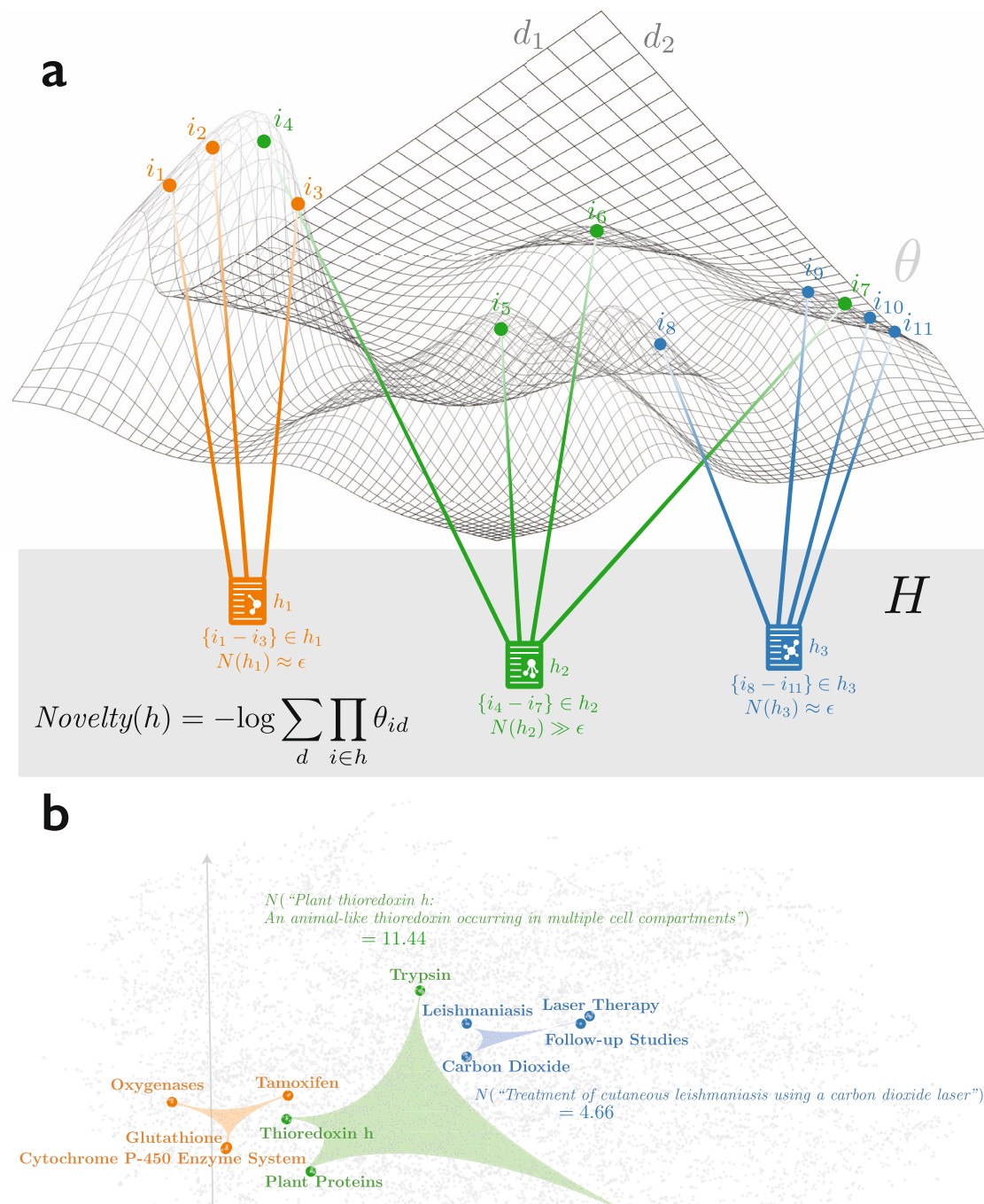

**Fig. 1 | Illustration of the embedding space and example combinations.**
**a** Illustration of the manifold inscribing all embeddings $\theta$ and an evaluation of three articles or patents (hyperedges $h_{1-3}$) in terms of their surprising combinations. Articles/patents $h_1$ and $h_3$ represent projects that combine scientific or technical components near one another in $\theta$, making each of high probability and low ($\epsilon$) surprise−similar to many related papers from the past. By contrast, paper $h_2$ draws a novel combination of components unlike any paper from the past, making it of

low probability and high ($\gg\epsilon$) surprise. See Supplementary Fig. 2 for a real density plot of the MeSH terms. **b**, Actual three-dimensional projection of the embeddings of a sample of MeSH codes from MEDLINE articles in our analysis. Also included are MeSH terms in the most surprising article (combination in the middle), the least surprising article (combination on the left), and a random article in between (combination on the right) among this sample of articles including four MeSH terms.

significantly with citation decile, as shown in Fig. 2a, b. The same pattern is observed in physical science (APS) papers and patents (Supplementary Figs. 4 and 5). Further, Nobel prize-winning papers, which are all in the top 10% citation group, have lower-than-average

context novelty, but high content novelty. Most general award-winning papers across Biology and Medicine lie in the top 10% of citations and follow the same pattern. This divergence between citations and awards for papers with high context novelty follows from the

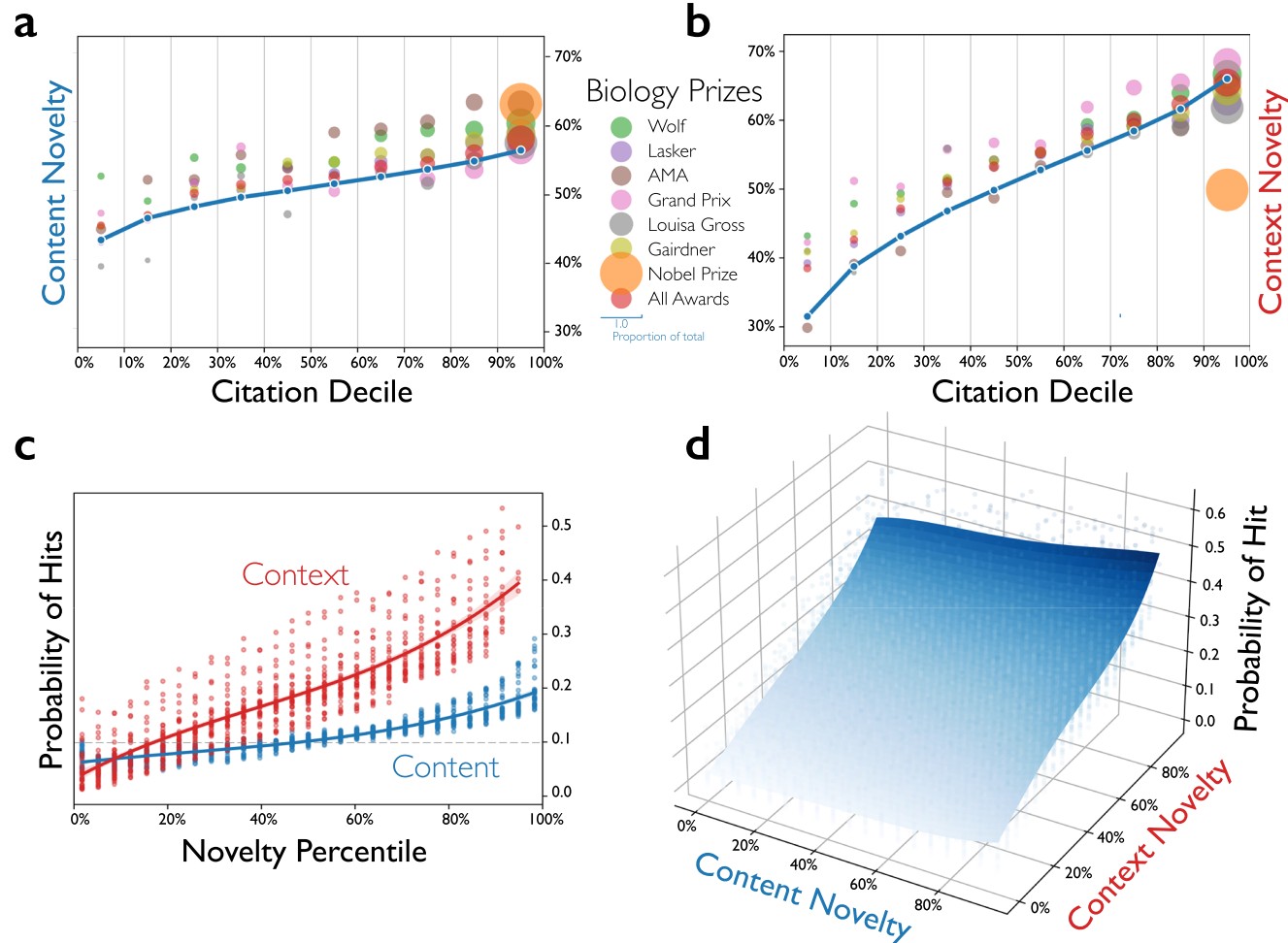

**Fig. 2 | Association between surprise and citation impact or awards for MED-LINE papers.** Mean content (**a**) and context (**b**) novelty for papers within each citation decile are plotted against the deciles, tracing a monotonic rise; Including averages for papers of Nobel prizes in Physiology or Medicine and general awards in Biology and Medicine. A piece-wise linear line connecting the data points is shown in each plot, with an error bar of 1 SEM around each data point which is too small to be observable. Probability of being a hit paper is plotted against content and context novelty separately (**c**) and jointly (**d**), manifesting a monotonic increase with novelty. Each dot in **c** represents a (novelty percentile, hit probability) data point for a certain year, and a third-order regression line to the data points is shown with a 95% bootstrap confidence interval. Source data are provided as a Source Data file.

distinction in audience. Citations are conferred by all scientists who credit an advance, but awards are offered by disciplinary communities, which overvalue advances within their contexts and undervalue trans-disciplinary research that violates context boundaries[43].

The probability of being a hit paper—in the top 10% of most cited papers published in the same year—increases monotonically with novelty percentile. For biomedical (MEDLINE) papers, those with the most surprising combinations of context are on average four times more likely to be a hit paper than random, surprising content combinations are two times more likely (Fig. 2c), and the most surprising content and context combinations are approximately five times more likely (Fig. 2d). These effects are amplified for super hits in the top 1% most cited papers (see Supplementary Fig. 6 and SI.) On average, nearly 50% of papers in the group of highest joint content and context novelties are hit papers, as shown in Fig. 2d. This predictive power outperforms other predictive models of hit papers in the literature. Moreover, it improves upon baseline models that predict impact with features—like content and context—available prior to publication (See SI for a detailed comparison.).

**Content versus context novelty**
Both content and context novelties predict outsized impact, but they provide nearly independent information regarding the ongoing

construction of scientific ideas and technological artifacts. The cor-relation between propensities for content and context combinations are low across biomedical science (MEDLINE: Pearson-$r$(5259751) <0.001, two-tailed $p = 0.94$), physical science (APS: Pearson-$r$(136274) = 0.03, two-tailed $p < 0.001$), and invention (USPatent: Pearson-$r$(2019493)<0.001, two-tailed $p = 0.29$). These findings sug-gest that our separation of content and context is necessary as they capture distinctive aspects of a scientific or technical advance. The experienced distinction between content and context novelty is rein-forced by expert classifications of biomedical papers from the Faculty Opinions platform, in which selected publishing scientists post favor-ite papers (74% of which are hits) and annotate them with predefined tags including "New finding", "New drug target", "Technical advance", "Interesting hypothesis", and "Controversial". When we consider papers selected and tagged by experts between 1990 and 2000, comparing against each other and all published papers from that period, we find that content novelty is most strongly and distinctively associated with New findings and Drug targets, while context novelty is most distinctively linked to Controversial, Interesting hypothesis and Technical advance (see Supplementary Fig. 7 and SI for details). These associations are above baseline, and distinct from one another. These associations provide support for our argument that novel content combinations yield legible advances within research fields,

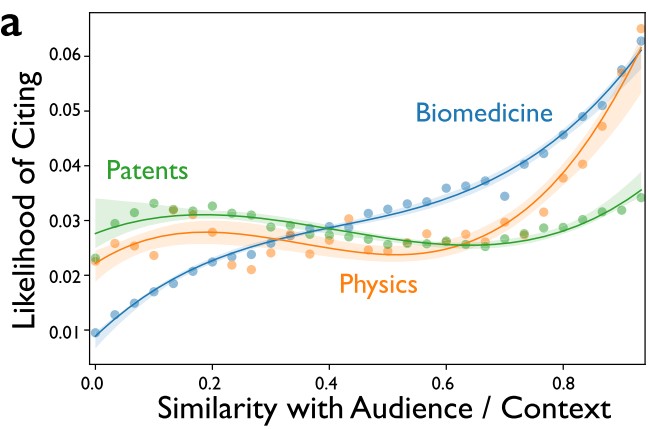
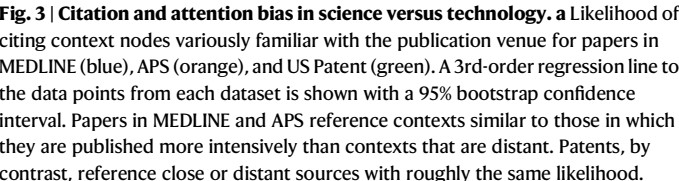
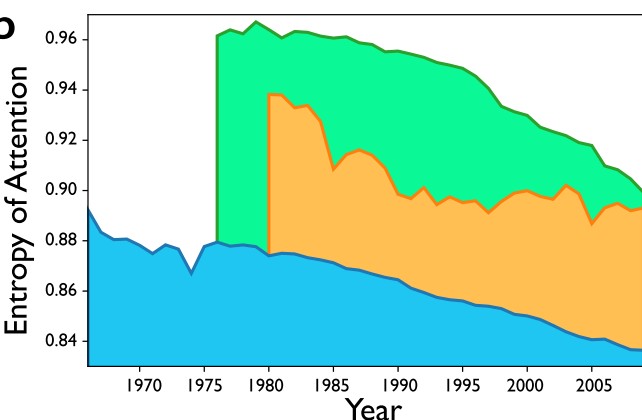

**Fig. 3 | Citation and attention bias in science versus technology. a** Likelihood of citing context nodes variously familiar with the publication venue for papers in MEDLINE (blue), APS (orange), and US Patent (green). A 3rd-order regression line to the data points from each dataset is shown with a 95% bootstrap confidence interval. Papers in MEDLINE and APS reference contexts similar to those in which they are published more intensively than contexts that are distant. Patents, by contrast, reference close or distant sources with roughly the same likelihood.

**b** Entropy of attention on the content nodes over time. The entropy of attention is calculated as the entropy of the number of publications associated with each content node. To compare entropy across years and datasets, it is normalized by the logarithm of the number of content nodes each year in each dataset. The content nodes in the patent space receive more equal attention (higher entropy), compared to MEDLINE and APS, across the years shown in the figure. Source data are provided as a Source Data file.

while novel context combinations generate field-violating controversy and surprise.

## Distinctions between science and technology

Results for patents relate to those from the biological and physical sciences in ways that illuminate distinctions between institutions of scientific and technological advance. The most surprising patents are two times more likely to be hits than random (Supplementary Fig. 5), but context novelty possesses less predictive power for patents. Disciplinary boundaries are weaker in technology space, where patent examiners, unlike scientific reviewers, do not enforce them. The lack of discrete fields enables inventors to search more widely, but removes the signal from violations of context in the prediction of advance. When we calculate content similarity between cited journals and venues where citing papers are published (see "Methods" for details), we see that scientists cite contexts similar to their publishing venue 500% more intensively than contexts that are distant (Fig. 3a)[44], presenting their work as if "standing on the shoulders" of likely reviewers[45]. Inventors of patented technologies cite distant sources with at least the same likelihood as those nearby. They are not reviewed by peers and present their work as if to highlight their novel distinction from neighboring work. Following from this difference, we find that the distribution of collective attention differs in science versus technology. We quantify the spread of attention with the normalized entropy of the number of publications containing each content node, shown in Fig. 3b. Keywords in the patent space receive more equal attention (higher entropy), while attention is more peaked and focused in biological and physical science.

## Sources of innovation

Finally, we assess sources of surprising advance in the form of surprising researchers, surprising research teams, and surprising research expeditions. A surprising scientist or inventor is one with an unexpected biography combining diverse research experiences. A surprising team is one comprising an unexpected combination of team members—scientists or engineers from an unusual collection of backgrounds. A surprising research expedition is one in which scientists or inventors travel from their disciplines an unexpected distance or direction to address problems framed by a distant audience. Using context embeddings, inscribed by journals and conferences, and Eq. 2, we quantify (1) career novelty for a scientist or engineer as the

improbability or surprisal of the combination of contexts in which she has ever published, (2) team novelty by the surprisal of contexts brought together across team members' publication histories, and (3) expedition novelty by the average distance between team members' publishing backgrounds and the publication venue of their focal paper or patent (See "Methods" for formal definitions). Figure 4a shows that for biomedical (MEDLINE) papers the probability of being a hit paper increases gradually with career and team novelty, but expedition novelty rises more quickly as the strongest predictor.

Papers representing the most surprising publication expeditions are 3.5 times more likely than random to be hit papers. Three-dimensional novelty distributions graphed in Fig. 4b also show that career and team novelties are correlated, suggesting that successful teams not only have members from multiple disciplines, but also members with diverse backgrounds who stitch interdisciplinary teams together. Successful knowledge expeditions (i.e., those that result in publications), however, are most correlated with breakthrough discovery. Figure 4b reveals how high expedition novelty in the absence of team and career novelty remains associated with an increased probability of hit papers. The pattern of effects in the physical sciences (APS) is consistent with those in the biomedical sciences (MEDLINE) with the association between expedition novelty and the probability of hit papers even more pronounced (Supplementary Fig. 8). These findings align with observations in the literature that creative researchers including Nobel laureates search across structural holes[46,47], as well as literature on individual and team impacts[48–52]. This pattern is different for patents (Supplementary Fig. 9) where expedition novelty is not significantly associated with hits because (1) novelty is the primary basis of patent evaluation, (2) subfields are not enforced and, as a result, (3) expedition novelty is so frequent that it loses its value as a signal of outsized impact: its skewness is .61, nearly three times larger than in the life sciences (skewness 0.26) and the flipside of physics, where such expeditions are less common (−0.36).

## Discussion

We offer support for Pierce's contention that abduction characterizes advance by showing that surprise anticipates disproportionate impact in science and technology. Up to 50% of outsized impact can be predicted by improbability under models that predict new research products. Moreover, we demonstrate how abduction is a collective process, typically occurring across teams of scientists and inventors

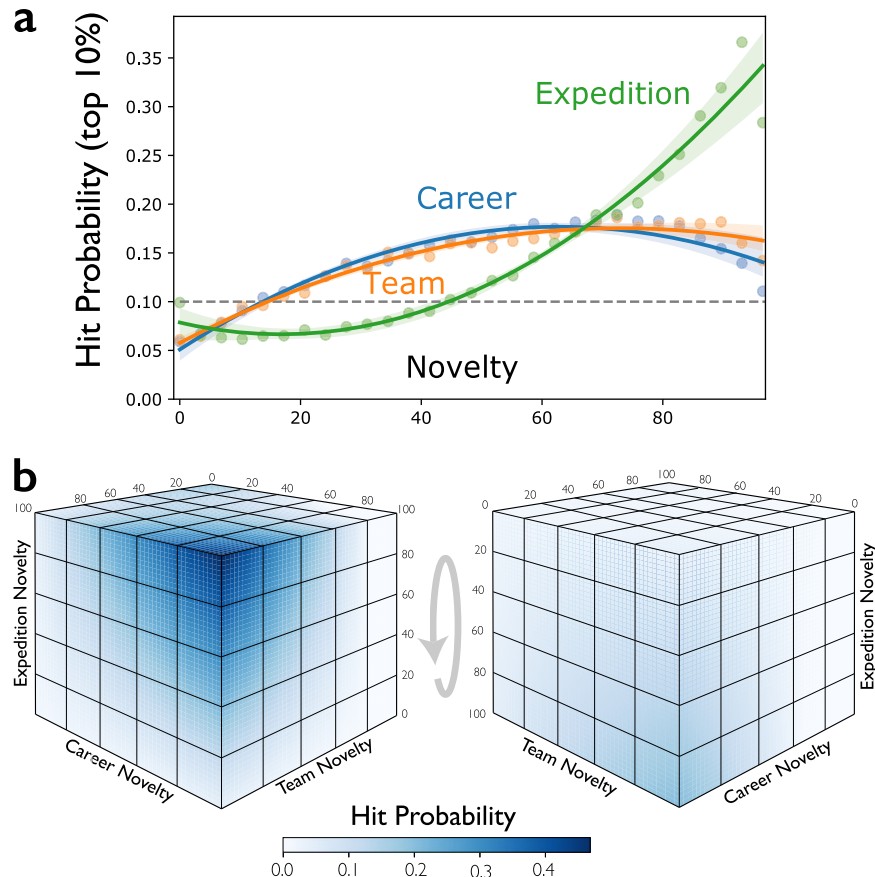

**Fig. 4 | Association between scientists' backgrounds and impact. a** Probability that a hit biomedical paper was produced by scientists manifesting greater career, team and expedition novelty; with career and team novelty closely correlated and expedition novelty sharply deviating. A 3rd-order regression line to the data points from each novelty type is shown with a 95% bootstrap confidence interval. **b** Hit probability as a function of career, team, and expedition novelties jointly with hit probability denoted by color. See "Methods" for details on how the heatmaps are produced. Source data are provided as a Source Data file.

rather than within them. The most surprising successes occur not through interdisciplinary careers or multi-disciplinary teams[53,54], but expeditions of scientists from one disciplinary context traveling to another. This implies that abduction is routinely social, where scientists from distant fields achieve substantial impact in advancing on a topic or challenge by bringing them into conversation with alien insights and perspectives.

We note several limitations associated with our study. In our analysis, we only evaluate our surprise measure on papers that pass the filter of peer review. We would like to test our model and measures on papers that did not make it through that process—presumably some containing speculative associations culled in review. Unpublished historical papers in most fields remain inaccessible, and we defer their investigation for future research, but we expect that the impact we find associated with surprise hinges on the collective skepticism novel work endures through peer review[55]. Another limitation follows from our coarse-grained operationalization of paper contents as keywords and contexts as cited journals. Like unpublished papers, full-text historical papers are unevenly available, and high-quality information extraction (e.g., equations, chemicals) from scanned technical journals poses a continuing challenge. Moreover, our modeling task distinguishes realized papers from random ones, rather than registering accuracy at predicting all published papers. Finally, our representation of each paper as a hyperedge—an unstructured bag of keywords—ignores the structural recipe at the heart of its scientific contribution. Despite these limitations, our analysis provides a model and measure of surprise that predicts some part of how scientists evaluate research articles and use them in future science.

Combinatorial frameworks have been a promising approach to conceptualizing novelty[17]. Here we operationalize this with an generative model based on hypergraph embedding and empirically demonstrate the power of higher-order structures in predicting future publications and patents that exceeds prior prediction efforts and even advanced deep learning architectures in capturing novelty and impact. By showing the generative sufficiency of the combinatorial approach for identifying next years' keyword combinations underlying published discoveries and patented inventions, we demonstrate that it is not simply one among many ways to conceptualize and model novelty, but the preferred model from which to anticipate advance. By revealing that the contents and contexts of research manifest a negligible correlation (<0.1), and that they map onto different scientific judgments from experts, this study disentangles and calls attention to two conceptually distinct dimensions of a novel contribution that had formerly been understood and measured as one. Moreover, while several have demonstrated a bias against novelty for receiving grants or achieving publication[9,56,57], we show how this is generally replaced by an impact bias favoring novelty among papers selected for successful publication.

Our findings suggest how models that predict expected and surprising advance represent tools for evaluating the degree to which scientific and technical institutions facilitate progress. For example, our results indicate that granting scientific awards for breakthrough progress, from Nobel Prizes to the plaques and certificates sponsored by nearly every scientific society, tend to reward conservative surprises. Scientific societies convene conferences, publish journals, and honor surprising combinations of scientific contents already familiar

to their members, even though surprising context combinations are more predictive of outsized citations and apparent scientific influence. This suggests that awards may represent a conservative influence on scientific advance[43,58,59]. Similarly, our findings reveal that scientists amplify the familiarity of their work to colleagues, editors, and reviewers, increasing their references to familiar sources by nearly 500%, likely in order to gain favor and appear relevant to reviewers and readers—to build on the shoulders of their audience[60–64]. This reinforces the internal focus of dense fields, which collectively learn more about less. Inventors, by contrast, cite and search widely to know less about more[65], providing new evidence for complementarities between search in science and technology and justifying why fundamental insights emerge not only from fundamental but also practical investigations[66–68].

Finally, in exploring collective abduction, our work formalizes the concept of a 'knowledge expedition', where scientists from one intellectual region travel to a distant other and address a problem familiar to the new audience, but with a surprising approach. These expeditions are distinct from interdisciplinary scientists or multidisciplinary teams, and if successful, they increase the likelihood that their work will disrupt the frontier and refocus scientific attention. This insight lays out a research agenda to understand the nature and influence of trans-disciplinary research expeditions. This becomes important in the wake of recent research demonstrating that scientific subfields may defend their internal approaches and understanding against invasion from outsiders[69,70]. Moreover, our findings call for analysis and experiments to explore the causal relationship between cross-disciplinary expeditions and punctuated advance. If expeditions do prove to systematically cause such advance, trans-disciplinary ventures could be systematically measured[71,72], catalyzed[73], incentivized[74] and expanded. The suggestive importance of expeditions for impactful surprise contrasts with research that focuses on inter- and multidisciplinarity as primary sources of advance[75–77], suggesting that they are also consequences of it. Furthermore, a causal relationship between expeditions and advance would hold implications for graduate education, which might be redesigned to target scientific and technological breakthroughs by sponsoring the trans-disciplinary search for problems, framing every student, team and expedition as an experiment, whose complex combination of backgrounds could condition novel hypotheses with the potential not only to succeed or fail, but radically alter the knowledge they conceive and the technology they create.

## Methods

There are no human or animal subjects involved in this study. According to the University of Chicago Social and Behavioral Sciences Institutional Review Board, the study "does not meet the definition of regulated human subjects research requiring review per 45 CFR 46.102(e)(1)."

### Data

This work investigates three major corpora of scientific and technological knowledge: 19,916,562 papers published between 1865 and 2009 in biomedical sciences from the MEDLINE database, 541,448 papers published between 1893 and 2013 in physical sciences from all journals published by the American Physical Society, and 6,488,262 patents granted between 1979 and 2017 from the US Patent database (USPTO). The building blocks of content for those articles and patents are identified using community-curated ontologies—Medical Subject Heading (MeSH) terms for MEDLINE, the Physics and Astronomy Classification Scheme (PACS) codes for APS, and United States Patent Classification (USPC) codes for patents. Then we build hypergraphs of content where each node represents a code from the ontologies and each hyperedge corresponds to a paper or patent that realizes a combination of the nodes.

We acknowledge the potential conservative influence from using established keyword ontologies rather than all of the words from titles, abstracts or full-text of articles and patents. Nevertheless, we note that the ontologies we examine do evolve over time, with active additions following the concentration of research in a given area. Moreover, these ontologies allow us to use the community of authors (APS), annotators (MEDLINE), and examiners (USPTO) to crowdsource the disambiguation of scientific and technological terms. Future work may explore how words differ from keywords, especially in the emergence of new fields.

**Medline.** MEDLINE is the U.S. National Library of Medicine's (NLM) bibliographic database. It contains abstracts, citations, and other metadata for more than 25 million journal articles in biomedicine and health, broadly defined to encompass those areas of the life sciences, behavioral sciences, chemical sciences, and bioengineering. The version of data used in this study contains 19,916,562 papers published between 1865 and 2009. Because coverage for papers prior to 1966 is limited, our main analysis focuses on papers published in and after 1966, but with pre-1966 papers as background information when predicting new content and context combinations. To allow the published papers to accumulate enough citations for assessing their impact, our novelty analysis focuses on papers published in and before 2000. In sum, all papers are used to estimate the hypergraph embedding model and 10,057,935 papers are used in the novelty analysis.

Medical Subject Headings (MeSH) is the National Library of Medicine's controlled terminology used for indexing articles in MEDLINE. It is designed to facilitate the determination of subject content in the biomedical literature. MeSH terms are organized hierarchically as a tree with the top-level terms (called headings) corresponding to major branches such as "Diseases" and "Chemicals and Drugs", with multiple levels under each branch. Terms in the bottom level are the most fine-grained, detailed concepts associated with distinct biological phenomena, chemicals, and methods. We use the bottom-level terms from the three branches that are central to the biomedical field— "Diseases", "Chemicals and Drugs", and "Analytical, Diagnostic and Therapeutic Techniques and Equipment" (or methods for short)—as nodes in the hypergraphs of content of MEDLINE papers. Terms from the Diseases branch include conditions such as "lathyrism" and "endometriosis"; examples from the Chemicals and Drugs branch include "elastin", "tropoelastin", "aminocaproates", "aminocaproic acids", "amino acids", "aminoacetonitrile", and "amyloid beta-protein"; and examples from the Analytical, Diagnostic and Therapeutic Techniques and Equipment branch include "polyacrylamide gel electrophoresis", "ion exchange chromatography", and "ultracentrifugation". NLM curators manually affix MeSH codes to papers as they are ingested into MEDLINE and made available through the popular PubMed platform.

**APS.** The APS dataset is released by the American Physical Society (APS). It contains 541,448 papers published between 1893 and 2013 in 12 physics journals: *Physical Review*, *Physical Review A, B, C, D, E, I* and *X*, *Physical Review Special Topics - Acceler and Physics*, *Physical Review Letters*, and *Reviews of Modern Physics*.

The dataset contains basic metadata for each paper including title, publication year, abstract, etc. It also contains the Physics and Astronomy Classification Scheme (PACS) codes associated with each paper. We use the PACS codes as nodes in hypergraphs of content to characterize APS papers. Similar to MeSH, PACS is also a hierarchical partition of the whole spectrum of subject matter in physics, astronomy, and related sciences. Example PACS codes include "Mathematical methods in physics", which range from "Quantum Monte Carlo Methods" to "Fourier analysis"; "Instruments…" such as "Electron and ion spectrometers" and "X-ray microscopes"; "Specific theories…" like

"Quark-gluon plasma" and "Chiral Langrangians"; and "...specific particles" ranging from "Baryons" to "Quarks". Unlike MeSH codes, which are added by curators, PACS codes are affixed by authors to their own papers through the publishing process. The PACS codes were developed by the American Institute of Physics in 1970 and have been used by APS since 1975. Since PACS codes are not available for papers published before 1975 and the coverage of them in papers prior to 1980 is limited, our main analysis is restricted to APS papers published after 1980, but with the pre-1980 papers as background information when predicting new content and context combinations. To allow the published papers to accumulate enough citations for assessing their impact, our novelty analysis focuses on papers published in and before 2000. In sum, all papers are used to estimate the hypergraph embedding model and 148,786 papers are used in the novelty analysis.

The dataset only contains citations between the APS papers. In order to obtain external citations we query the Web of Science (WOS) database to collect all the journals cited by the APS papers. Particularly, in the WOS database we find all the papers published by the 12 APS journals, and then all the journals cited by those papers. The journals are then used as nodes in hypergraphs of context for the APS papers. In addition, we query the WOS database to collect the number of citations a paper receives for more accurate assessment of the paper's impact.

**US Patents.** The US Patent dataset is released by the US Patent & Trademark Office (USPTO). It contains 6,488,262 patents published between 1976 and 2015. The dataset contains basic metadata for each patent such as title, publication year, USPC (United States Patent Classification) codes, etc. The USPC is a classification system used by USPTO to organize all U.S. patent documents and other technical documents into specific technology groupings based on common subject matter. The USPC is a two-layer classification system. The top layer consists of terms called classes, and each class contains subcomponents called subclasses. According to USPTO, a class generally delineates one technology from another and every patent is assigned a main class. As such, we use the class codes as nodes in the hypergraphs of context for patents. Subclasses delineate processes, structural features, and functional features of the subject matter encompassed within the scope of a class, and thus we use subclass codes as content nodes for the patents. In total, there are 158,073 subclass codes (content nodes) and 496 class codes (context nodes).

To allow the granted patents to accumulate enough citations for assessing their impact, our novelty analysis focuses on patents granted in and before 2000. In sum, all patents are used to estimate the hypergraph embedding model and 2,436,257 patents between 1976 and 2000 are used in the novelty analysis.

**Nobel prize papers.** The Nobel prize-winning papers are derived from the Nobel laureates dataset by Li et al.[78], which contains publication histories of nearly all Nobel prize winners from the past century. However, the focus of that dataset is on the Nobel laureates, but our study focuses on award-winning papers. While it is relatively easy to find out the person who won a prize, it is hard to pinpoint the papers that contribute to the winning of the prize. Li et al. take a generous approach by including papers cited by Nobel lectures and papers published in the same period of one's prize-winning work (while satisfying several inclusion criteria). This results in noise for our analysis as not every paper in their dataset is a prize-winning paper. As a conservative solution, for every Nobel laureate we take their most cited paper in the dataset and use only those papers as award-winning papers in our analysis. We acknowledge that a few Nobel prizes are attributed to an opus of work and this filtering process might miss a few relevant papers, but the most important (in terms of impact) paper for every prize is kept, with every paper representing an award-winning paper.

**General award-winning papers.** In Fig. 2, we compare the content and context novelty of our entire population of MEDLINE papers with those receiving awards. The general awards data is from Foster et al.[25]. They define award-winning papers as those authored by a scientist who won an international award or prize. They first created a large list of prizes by drawing from the category pages for biology awards, medicine awards, and chemistry awards in Wikipedia and then validated them with several biomedical scientists. Then they identified the winners of 137 different prizes and awards from biology, medicine, and chemistry. For each winner, if at all possible, they retrieved all papers written from 0 year to 30 years before the award was granted.

False positives could be introduced due to mis-assignments of papers to awards. However, as these false-positive papers would be typical rather than award-winning, they likely behave like the majority of published papers and dilute the effect we report. False negatives are also possible if we missed papers published by certain award winners. For instance, papers written by authors with non-English characters in their names are underrepresented. These false negatives similarly dilute our estimate of the distinctiveness of award-winning papers. Our non-Nobel prize findings should therefore be viewed as a conservative estimate of the difference between prize winning papers and the pool of all papers. Still, we are able to identify that prize winning papers are significantly higher in content novelty than the average paper within a given citation decile, but that they are not systematically higher in context novelty, likely because awards are typically conferred by a single context—a journal, association, or field.

## Hypergraph embedding model

We model discoveries and inventions as hypergraphs of contents and contexts with a combinatorial process that assembles previous concepts and technologies from prior papers or patents. Past research examining combinatorial discovery and invention has modeled only partial combinations, deconstructing new products into component pairs[8,79] and resting on mature analysis tools for simple graphs, which define direct links between entities. Here we model science and technology as a complex hypergraph drawn from an embedding of research contents and contexts, where individual discoveries or inventions are rendered as complete sets of each. For a given hypergraph, whether composed of content or context nodes, the propensity of any combination of nodes to form a hyperedge is modeled as a product of two factors: proximity between nodes in the combination and their cognitive salience. Combinations with higher propensity will be more likely to turn into papers and patents, agreeing with the intuition that people tend to search locally and pursue trending topics.

To formalize this idea, each node $i$ is associated with a latent vector $\theta_i$ that embeds the node in a latent space constructed to optimize the likelihood of observed papers and patents. Each entry $\theta_{id}$ of the latent vector denotes the probability that node $i$ belongs to a latent dimension $d$, and thus $\sum_{d=1}^{D} \theta_{id} = 1$. Dimensions underlying these latent vectors naturally recover scientific fields; see Supplementary Fig. 1. The complementarity between nodes in a combination $h$ is modeled as the probability that those nodes belong to the same dimension, $\sum_d \prod_{i \in h} \theta_{id}$. This hypergraph embedding model can be understood as an extension of the mixed-membership stochastic block model[32], which was designed for networks with only pairwise interactions.

We also account for each node's cognitive availability because most empirical networks display great heterogeneity in node connectivity, with a few popular contents and contexts attracting the most attention—intensively drawn upon by many papers and patents. Previous work has shown that by integrating heterogeneity of node connectivity, the performance of community detection in real-world networks improves[80]. Accordingly, we associate each node $i$ with a latent scalar $r_i$ to account for its salience, presumably tied with its overall connectivity in the network.

Assembling these components, we model the propensity ($\lambda_h$) of combination $h$—our expectation of its appearance in actual papers and patents—as the product of proximity between nodes in $h$ and their salience or visibility:

$$\lambda_h = \sum_d \prod_{i \in h} \theta_{id} \times \prod_{i \in h} r_i.$$

Then the number of publications or patents that realize combination $h$ is modeled as a Poisson random variable with $\lambda_h$, the propensity of that combination, as its mean:

$$X_h \sim Poisson(\lambda_h)$$

Accordingly, the probability of observing a hypergraph $G$ is the product of probabilities for observing all possible combinations:

$$P(G|\Theta,R) = \prod_{h \in H} P(x_h|\Theta,R),$$

where $x_h$ is the number of observed papers or patents that realize combination $h$ and $H$ is the set of all possible combinations. $(\Theta,R)$ denotes all unknown parameters:

$$\Theta = (\theta_1,...,\theta_n) \text{ and } R = (r_1,...,r_n).$$

Finally, we model a time sequence of hypergraphs $\left(G^1,...,G^T\right)$ as the output of a Hidden Markov Process on latent parameters $\Theta,R$:

$$P\left(G^1,...,G^T|\Theta^1,...,\Theta^T,R^1,...,R^T\right) = P\left(G^1|\Theta^1,R^1\right)$$
$$\prod_{t=2}^{T} P\left(\Theta^t,R^t|\Theta^{t-1},R^{t-1}\right) P\left(G^t|\Theta^t,R^t\right),$$

where time is indexed by the superscript $t$, and the transition density $P\left(\Theta^t,R^t|\Theta^{t-1},R^{t-1}\right)$ is a Gaussian density.

Given articles published by a certain year $T$, we estimate parameters $\left(\Theta^1,...,\Theta^T,R^1,...,R^T\right)$ by maximizing the likelihood function above via stochastic gradient descent. Then the model enables us to predict combinations in year $T + 1$. However, even with stochastic gradient descent, model estimation is still computationally challenging due to the vast space of possible combinations. We address issues in the estimation process as follows.

First, the space of possible combinations is exponentially large (on the order of $2^n$), and it is computationally prohibitive to go over all possible combinations even with stochastic gradient descent. However, it is rare for large combinations to turn into hyperedges, and hence, we restrict the set of possible combinations to include only combinations no larger than the largest hyperedge observed.

Second, because the real hypergraphs are sparse, the sets of hyperedges and non-hyperedge combinations are exceedingly unbalanced with the number of hyperedges to be on the order of $n$ but the number of non-hyperedge combinations on the order of $n^D$ (where $D$ is the size of the largest hyperedge). We employ a widely used approach in machine learning, negative sampling, to address this imbalance issue. Specifically, in each iteration of the training (optimization) process, we randomly sample as many non-hyperedge combinations as hyperedges to construct balanced hyperedge and non-hyperedge sets. Negative sampling effectively changes the objective function we optimize. The original objective function can be rewritten as $\log P(G|\theta,R) = \sum_{h \in E} \log P(x_h|\theta,R) + \sum_{h \in \bar{E}} \log P(x_h|\theta,R)$, where $E$ is the set of hyperedges and $\bar{E}$ is the set of non-hyperedge combinations. Negative sampling turns the second term in the objective function into a random function $X = \sum_{h \sim P(h)} \log P(x_h|\theta,R)$ where $h$ is randomly drawn from $\bar{E}$. This random function is a biased estimator of the

original objective. To illustrate the bias, assuming the simplest case of stochastic gradient descent where we draw 1 negative sample for each positive sample, the expectation of $X$ then becomes $\sum_h \log P(x_h|\theta,R)P(h) = \frac{1}{|\bar{E}|}\sum \log P(x_h|\theta,R)$ where $|\bar{E}|$ is the size of $\bar{E}$ and $P(h) = 1/|\bar{E}|$ because $h$ is a draw from $\bar{E}$ uniformly at random. The result for more complex stochastic gradient descent algorithms can be derived similarly with more complicated factors in front of the second term. In general, the effect from the second term is down-weighted due to negative sampling. Despite this estimator's statistical bias, it works well in practice for word embeddings[81], question answering[82], and many other applications, outperforming the comparable but unbiased contrastive learning approach by co-minimizing bias and variance[81]. We also found this approach worked best in our model as seen in high predictive power despite bias. Nonetheless, we acknowledge that estimation could be further improved by considering more advanced sampling schemes[83], which we defer to future work.

Lastly, to facilitate the stochastic gradient descent, we take a mini-batch of hyperedges and non-hyperedges to compute the gradient of the objective function at each step of the training process.

## Model evaluation
As a brief summary, we investigate three datasets: MEDLINE, APS, and US Patent; each dataset contains hypergraph data over several decades; and we model content and context hypergraphs separately. Consequently, for each dataset, for each year $t$ covered by the dataset, we fit the model to the hypergraph of contents and separately to the hypergraph of contexts up to year $t$. In total, we estimate $2 \times (T_{MEDLINE} + T_{APS} + T_{Patents})$ models where $T$ is the number of years covered by our study from the corresponding dataset. Then, we evaluate the fitness of each model by its predictive performance of (out-of-sample) future combinations.

For example, given hypergraphs of MeSH terms up to and including year 2008, we estimate the hypergraph embedding model, and use the estimated model to predict hyperedges in 2009. Specifically, using the estimates of the parameters $(\theta,r)$, we can compute the propensity $\lambda_h$ of any combination $h$ of MeSH terms in year 2009, following Equation (1) $\lambda_h = \sum_d \prod_{i \in h} \theta_{id} \times \prod_{i \in h} r_i$. Then we assess the model's predictive performance in terms of its AUC (Area Under the Operator-Receiver Curve). Statistically, AUC is the probability that a random combination which turned into a hyperedge (positive combination) in 2009 would have a larger propensity than a random combination that did not turn into a hyperedge (negative combination) in 2009. To estimate this quantity, we randomly sample a positive combination and a negative combination of the same size from 2009, and check whether the positive combination has a larger propensity than the negative. The simulation is repeated for 10000 times and we calculate the fraction of times where the positive has larger propensity than the negative, which is the estimation of the AUC score in predicting hyperedges in 2009. It is easy to see that a perfect predictor would achieve an AUC score of 1 and random guesses would have an AUC of around 0.5. The larger the number, the better the predictive performance.

## Probability of hit papers and patents
A hit paper (or patent) is defined as one among the top 10% most cited papers (or patents) published in the same year. For example, for all the papers published in 1990, we count all the citations they received in the time period covered by our dataset, and the top 10% most cited papers are hit papers.

To study the effect of novelty on the probability of being a hit paper, for all the papers published in each year, we first transform the raw novelty scores of the papers into percentiles, and then divide them into 30 equally sized bins in ascending order of the novelty scores. Then we assess the probability of hit papers for each bin as the fraction of hit papers in that bin. Finally, the probability of hit papers for each

bin is plotted against the center percentile value in that bin in Figs. 2, 4, Supplementary Figs. 4, 5, 6, 8 and 9.

Similarly, to study the joint correlation of different novelties with the probability of being a hit paper, we divide the papers into multiple bins according to content and context novelty simultaneously. For the joint effect of content and context novelty, we divide the paper into 30-by-30 bins in terms of both novelties, then calculate the fraction of hit papers in each bin. Next we regress the probability of being a hit to content and context novelties. Fitted values of hit probability are then visualized as a 2D heat map across the full range of content and context novelty scores.

For the joint effect of career, team, and expedition novelty, we divide the papers into 20-by-20-by-20 bins in terms of their source novelty scores, and then calculate the fraction of hit papers in each bin. Again, we fit a polynomial function between hit probability and the three novelties to obtain the average relationship:

$$P(hit) \sim \sum_{i,j,k=0}^{1} careernovelty^i \times teamnovelty^j \times expeditionnovelty^k$$

The resulting 3D heat cube comprises 6 2D heatmaps where color denotes the fitted probability of hit for any given values of (career novelty, team novelty, expedition novelty).

A detailed comparison with other models in predicting hit papers or patents is presented in the Supplementary Information.

## Preferences on context in citations

To assess the intensity with which scientists and inventors cite contexts (e.g., journals and conferences) familiar to their audience, we compute the similarity between every pair of context nodes where one cites the other. For example, for a paper $i$ published in journal $X$, we calculate the similarity between the journal $X$ and every journal cited by paper $i$. Similarity is quantified by the cosine similarity between two vectors representing the content of the two journals, conditioned on the content of paper $i$. Specifically, each journal is represented by a vector and each entry in the vector corresponds to a content node (MeSH terms, PACS codes, or patent subclasses); the value of an entry is the number of papers containing the corresponding content node and ever published by the journal, appropriately normalized so that the sum of the vector is 1. In other words, the vector consists of the loadings of the journal on different contents. When calculating the similarity between two journals, we do not directly compute the cosine similarity between their vectors, as those vectors contain substantial information irrelevant to the paper currently under consideration. Instead, we use only the entries corresponding to content nodes in paper $i$ to calculate the cosine similarity between the two journals.

As we sweep through all papers (or patents), we obtain a distribution of the similarity between citing-cited context pairs: the number of times for which context nodes at a given similarity with the audience context (i.e., the citing context) are cited. To normalize this distribution, we also compute the potential space of citation similarity—the number of times at which context nodes at a given similarity would be cited at random. We achieve this by the following procedure: for each paper, sample as many context nodes uniformly at random from all context nodes as those originally cited, treat the sampled context nodes as if they were cited by the paper, and carry out the same similarity calculations as above. Finally, we have two distributions of similarity between citing-cited context pairs—one observed and one simulated by random sampling—and we take the ratio of the two as the likelihood of citing a context at a given similarity with the audience's context.

## Career, team and expedition novelty

Assuming $\theta_i$ is the embedding of context node $i$ (e.g., a journal or conference), we define career, team, and expedition novelty as follows.

**Career novelty.** For any scientist, we collect all contexts (e.g., journals) in which she has ever published papers into list $C$. Then we calculate the surprisal across all contexts in her career using Eq. (2): $career\ novelty = -\log \sum_d \prod_{i \in C} \theta_{id}$. Next, given a paper, we use the average career novelty across the authors of the paper to assess how career novelty associates with paper impact.

**Team novelty.** For any paper, we first pool all the contexts (e.g., journals) where authors of the paper have published, without duplication. We denote this list of contexts by $T$. Next, we use Eq. (2) again to calculate the surprisal of this list: $team\ novelty = -\log \sum_d \prod_{i \in T} \theta_{id}$. Finally, we compare the paper's team novelty to its impact to assess their association.

**Expedition novelty.** For any paper, we again collect all contexts where its authors have published into a list $T$. Denoting the context where this paper is published by $\theta_v$, we calculate the expedition novelty of this paper (or the authors of the paper) by $expedition\ novelty = average_{i \in T}(1 - \theta_i \cdot \theta_v)$.

Our career, team and expedition novelty measures also contribute to the literature on interdisciplinarity measures[84], which include prominent measures that approach interdisciplinarity from the perspective of journal keyword combinations using the Web of Science[75,85,86] and diversity or inequality indices (e.g., the Simpson Index, Shannon Entropy, and the GINI coefficient) assessed over these keywords[87,88]. Our measures differ from those above by being more granular and embedding the keywords into a continuous, high-dimensional space. Other work in the literature accounts for the similarity or difference between research fields combined[89,90]. Our measure does this automatically by accounting for the combinations of keywords required to characterize current literature and predict future research. Still others are built atop measures of centrality in networks of publication. By measuring the inner product of term vectors, our measure also approaches a continuous measure of centrality within the embedding space of contents and contexts. In short, our measure captures desiderata from each of the major categories of novelty measurement—diversity of combination, accounting for differences between components, and which represents centrality in the continuous embedded manifold.

### Reporting summary

Further information on research design is available in the Nature Portfolio Reporting Summary linked to this article.

## Data availability

The raw MEDLINE data are available at the PubMed database (https://pubmed.ncbi.nlm.nih.gov/download/) and the processed MEDLINE data used in this study are open-access at Harvard Dataverse (https://doi.org/10.7910/DVN/NFSYYA). The APS data used in this study are available at https://journals.aps.org/datasets. These data can be obtained through APS by submitting a request. The US Patent data used in this study are open-access in the patentsview database at https://patentsview.org/download/data-download-tables. The "Nobel Prize Papers" data used in this study are open-access at Harvard Dataverse https://dataverse.harvard.edu/dataset.xhtml?persistentId=doi:10.7910/DVN/6NJ5RN. The "General Award-Winning Papers" data used in this study are open-access at Harvard Dataverse https://doi.org/10.7910/DVN/NFSYYA. The "facultyopinions" data used in this study are open-access at Harvard Dataverse (https://doi.org/10.7910/DVN/NFSYYA). Source data are provided with this paper.

## Code availability

Code that supports the main findings of this study are available on GitHub:[91] https://github.com/KnowledgeLab/hyper-novelty.

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

## Acknowledgements

We are grateful to workshops at MIT Sloan, the Santa Fe Institute, and the Computational Social Science program at the University of Chicago for helpful comments. F.S. was supported in part by the Data@Carolina initiative; F.S. and J.E. were supported by the John Templeton Foundation; J.E. was supported by the Air Force Office of Scientific Research (FA9550-19-1-0354), National Science Foundation (1829366), and DARPA (HR00111820006).

## Author contributions

F.S. and J.E. designed the research. J.E. obtained the MEDLINE, APS and faculty opinions datasets. F.S. obtained the patent dataset. F.S. implemented the model, and performed all analysis. F.S. and J.E. wrote the paper.

## Competing interests

The authors declare the following competing interests: pending patent application. Patent applicant: University of Chicago; inventors F.S., J.E. and Jamshid Sourati; US Patent Application No.: 17/451,320 for "Systems and methods for high-order modeling of predictive hypotheses." Specifically, this patent claims the use of our hypergraph prediction model (along with another model not described in this paper) for generating novel technology hypotheses.
