## [Peer Review File · Nature Communications]

Science and Technology Advance through Surprises Produced
by Expeditions of OutsidersREVIEWER COMMENTS

Reviewer #1 (Remarks to the Author):

This manuscript studies the relationships between "surprise" of scientific or technological discoveries and their success using a generative hypergraph model. The hypergraph is defined on the space of ontologies or classification systems (e.g., MeSH terms or patent subclass code) or venues (journals). The generative model defines, given the parameters (embedding of terms and the scalar reflecting the degree), a propensity function of possible hyperedges with a Poisson model (i.e., number of papers or patents with the given set of terms). The paper shows that the model can learn useful embedding from the past that can distinguish realized hyperedges from unrealized ones in the next period. The propensity function is computed as a product of "complementarity" and "availability". Each node is represented as a normalized latent vector. The complementarity is defined by examining whether all nodes in the combination are expressed in the same dimensions; the "cognitive availability", which roughly prescribes the degree of nodes, is modeled with a scalar parameter for each node (and their product for a hyperedge). Finally, the propensity of a hyperedge is computed as a product of these two factors. Two separate hypergraphs are considered to capture the "content" and "context". These graphs are separately defined for each dataset (e.g., MeSH terms vs. journals; PACS codes vs. journals; and subclass codes vs. class codes). Finally, novelty is defined as the surprisal of its complementarity.

With the novelty defined for the content and context, the authors showed that the novelty and citation are positively correlated. Namely, as the citation increases, the mean novelty also tends to increase (although content and context novelty are not strongly correlated). The probability of hits also increases with novelty. Finally, it was shown that the novelty measures defined for a team, career, and "expedition" capture some associations with the hit probability.

This paper proposes an intriguing, very clever hypergraph embedding model to capture knowledge space and its exploration. The combination of the hypergraph-based operationalization of knowledge products and the "complementary" formula that specifically captures the multi-dimensional combinatorial property seems like an innovative way to address the challenges of operationalizing novelty, moving beyond the existing combinatorial definitions of novelty. The model is validated by using a temporal link prediction task (using a model trained on the past to predict future hyperedges) and the result seems to indicate that the model indeed learns useful embeddings for predicting knowledge products. The results on the association between novelty and success are also fascinating. Therefore, I believe that this is an important contribution to the science of science.

At the same time, I am unsure about several results and their presentations and I think it is important to clarify them before recommending them for its publication.

- I'm guessing that the vast majority of new hyperedges overlap with the existing hyperedges from the prior years. Is this true? One can think of a simple baseline where we predict "Yes" for previously seen hyperedges and "No" for previously unseen hyperedges. How well can this baseline perform in the evaluation task? I think the performance of this baseline would provide a useful context to the performance of the model.

- Across all figures, whenever possible, I would suggest showing the uncertainty of the estimations. Without uncertainty, it is difficult to judge whether the pattern is real or not.

- In Fig. 2, (a) and (b) show only the average value per decile. I think there's a missed opportunity to reveal more information -- how they are related in more detail. What's the uncertainty here? How is the novelty distributed for each citation decile? What happens if you flip the x and y-axis? Given their novelty, do prize winners tend to have fewer citations? Have you examined the whole distribution (say, a heatmap of the citations similar to (d) or a 2D version of it)? Wouldn't this show useful details

about the distribution of novelty and citations?

- In Fig. 2, (c) and (d) shows many data points as dots, but I could not find what they are. I'm guessing that each dot may represent a bootstrapped sample but the details on these points couldn't be found.

- In Fig. 4, it is unclear how the "heat-cube" is colored. The authors should specify and justify the method to obtain the smoothed map.

- The main manuscript says, "MEDLINE papers with maximal joint context and content surprise predict nearly 50% of the likelihood of being in the top 10% of citations". I was not sure what exactly this 50% number means. I'd suggest elaborating a bit more on the task setup and how exactly this number was calculated.

- I think more elaboration on the concept and precise operationalization of 'content' and 'context' in the main text would improve the readability of the paper. Although reading the method section clarified for me, I was confused about how they are operationalized. A clearer, more explicit presentation of 'content' and 'context' (hypergraphs) may be helpful for the readers.

- I would encourage authors to consider rephrasing the claim that "the vast majority of new scientific and technological configurations can be predicted". Yes, it was shown that the model can distinguish the combinations of MeSH terms that happen from random ones, but I think the term "configuration" can mean many different aspects of knowledge products and thus the statement can be easily misinterpreted. Also, depending on the performance of the super simple baseline, this fact may potentially be much less surprising than it looks.

- The name "block model" is a misnomer. In block models, the probability of an edge (propensity) only depends on each node's block membership and that's why they are called block models. Here, the proposed model does not have 'blocks' that dictates the propensity. It is rather just a hypergraph embedding model, similar to the geometric embedding models (e.g., hyperbolic space embedding models, word embedding models, etc.). I think the current name would confuse those who are familiar with block models. Therefore, I'd strongly encourage the authors to change the name to avoid unnecessary confusion.

- The method uses negative sampling, which is a biased and simplified version of the Noise-Contrastive Estimation (Gutmann & Hyvärinen, 2010). Although this may not be a huge issue because word2vec and other models with negative sampling work fine, the authors may want to discuss the fact that the negative sampling is a biased estimator and there may be certain consequences due to this bias.

- Not much detail was provided regarding the details of model training and thus I don't think it is possible to replicate the results just based on the paper. The code may provide all necessary details, but the GitHub address is not disclosed in the paper. Could you specify hyperparameters? For instance, what are the hyperparameters for SGD? What are the largest hyperedge size for each dataset? What's the hyperparameters for the negative sampling?

- Right after Eq. 2 says, "We note that our deep learning-based transformer predictions of content and context ...". But I believe that the manuscript does not talk about the transformer model at all, but it is only mentioned in the SI. I think this paragraph is not necessary in the main manuscript.

- In Figure 1 (b), I think the green, not the blue, is the most novel article.

Reviewer #2 (Remarks to the Author):

I am pleased to submit a review of this very interesting paper, "Science and Technology Advance through Surprise, driven by Expeditions of Outsiders." The paper takes computation approach to search for patterns in large amounts of scientific articles and patents using existing ontologies for scientific fields and technology areas. The approach seeks to develop a model of abduction following C.S. Pierce.

(My copy did not have page numbers so I numbered the pages starting with the abstract on page 1.)

The paper's findings depend on the success of a model created for experimentation with a large corpus of scientific output including journals, articles, and patents. It is difficult for this reviewer to assess the propriety of the model. It would be difficult to draw from this paper what other possibilities for experimentation might have been possible. Is this the best model for assessing patterns - it appears to be an emergent model, although there is reference to higher order structuring (p.4). It is not clear at what level this paper is operating if one assumes a hierarchical model.

In most social science fields, scholars avoid using the term "predict" because of the existence of non-linearity and multivariate conditions, thus it concerns this reviewer to see the term 'predict' used a number of times in the paper. It seems as if the model is set up to search for what it finds, although I may not understand it fully.

The idea of a complex hypergraph of scientific advance is intriguing and appears to be an interesting approach, but, in the paper itself, the methodology section only briefly describes the process. As this reviewer reads it, the method seeks clusters of terms - is that correct? The clusters are derived from MeSH terms? The terms are placed into datasets by year to build a hypergraph for that year, although I am not totally clear on this method. Reading the more in-depth method appendix still leaves it unclear as to what variables are being examined, and how many different experiments were run. Can you please clarify the variables and experiments?

The reference in page 1 to serendipity of science is apt, but there have been many different papers and experiments delving into the dynamic structure of science beyond Merton. These works do not seek to predict what will happen in the future -- that is not the goal of any known research in sociology of science. However, the work of Leydesdorff, in particular, deals with the complex dynamics of communications structures such as self-organizing criticality of discovery and Shannon uncertainty in communication structures - and either of these concepts would be good to discuss in light of the approach suggested in this paper.

The contrast with patent data is interesting, but it might be better to have a separate paper for the patent data, and only refer to it briefly in this paper. The discussion does not apply directly to the focus on surprise in scientific advances. As the authors note on page 7, expectations for patents is quite different than for papers. Moreover, it is unclear how the patent discussion supports the initial goal of demonstrating abduction.

On page 3, the text reads: "The model implies..." this may be a small thing, but a model cannot imply something. The next sentence says that local search agrees with previous findings published by one of the authors. In fact, the concept of local search in scientific communities has a long history in sociology of science, so it might be better to cite other work preceding this paper rather than the author's own work.

I very much like the finding on page 5-6 on content and context differences in their relationship to outcomes. The paper does not set this up as a hypothesis - seeking only abduction - but the relationship between content, context and outcomes is an interesting insight. Can you say what the significance measures are for this finding? How does it relate back to the initial hypothesis?

The discussion of teams and expeditions is also very informative. The findings make some contributions to the science of teams. Here, I wonder what 'career novelty' and 'team novelty' consists of, since that is not defined, page 6. It has been shown that the work of teams has higher impact over time than the work of sole authors. The findings here seem to support that work. Perhaps a reference to that earlier work would be helpful here.

On page 7, the authors refer to collective abduction, which seems like an intriguing idea. It might be worth considering defining this term fully and explaining how the operation shows this feature. Similarly with the idea of 'collective attention.' Here, the paper also discusses exposure to foreign theories that enable surprising discoveries. Again, it would be good to have more information here. Several background papers might be worth consulting about surprise, including "Being Surprised and Surprising Ourselves," by Dragos Simandan (2018), and "Tools for Thought" by H. Rheingold (1985).

Also on page 7, the authors refer to cross-disciplinarity, interdisciplinarity, and multi-disciplinarity. Here, no doubt, the authors are aware of a deep literature on these subjects. Measures have been developed that could also be applied to assess changes in the knowledge base. A discussion of these methods as they are similar to or different from the method proposed here would be enlightening.

On page 8, the authors suggest that the method could be applied to institutions to help facilitate progress and also suggest that the method could help foster advances. It may be premature to suggest a normative application of this method. The model offers an intriguing way to study the structure of scientific advance, but it is early days in testing it; its possible applications lay in the future. Institutions generally do not want to have metrics of 'progress' put on their creative work.

Similarly, also on page 8, the finding on prizes going to conservative advances, and that people cite familiar sources, have both been shown in the sociology of science in several different contexts, so it might be good to cite these works.

The idea of knowledge expeditions (while not termed in this way exactly) also supports other work that has been done in the past. It is unclear why the authors see their findings as distinct from interdisciplinarity since no effort has been made in the paper to distinguish findings here from others. However, it has been noted that researchers search for new ideas and that creative researchers put together work across structural holes (Burt, 2002). It has also been shown that Nobel Prize winners span structural holes Wagner et al., 2015).

Small item, but footnote 36 does not appear in the references.

It was disappointing to get to the end of the paper and not find a discussion of how the model upholds or refutes the concept of abduction. This appeared at the beginning to be the purpose of the article, yet, there is no further discussion about this hypothesis in light of the workings of the model. The idea is intriguing, and it would be good to say more at the opening as to how the model will reveal abduction. The conclusion should say something about the hypothesis and the extent to which it reveals this dynamic.

Overall, there are many observations and findings in the paper that could fruitfully refer back to earlier theories, models, and concepts in the dynamics of scientific advance. Some of the results support earlier work and other points refute it. Scholarship would be greatly advanced by a discussion of how the model tests and challenges these earlier concepts. Moreover, many of the earlier works have been written by women, and thus it was disappointing to see so few women scholars cited in the references.

I look forward to seeing this work develop in the months and years ahead. Good luck with it.

Reviewer #3 (Remarks to the Author):

This paper examines the role of surprise in advancing scientific knowledge. It combines a series of methodologies from machine learning and statistics (e.g., embeddings, hypergraphs, stochastic block models, attention-based neural networks) using community-curated ontologies to investigate whether various content/context combinations and whether they can be predicted. (I wonder if this paper's unique combination of various methodologies will increase its probability of success?) There is a substantial amount of work that has been invested in this paper with large-scale data and complex analysis tools. Overall, I find the paper a nice contribution to science studies. It will generate conversations about surprise, discovery and novelty in science and technology. My main concern is the methods are a bit oversold for what they say about surprise and discovery in science. If the authors can tone this down a bit and can address some of my other concerns, I think the paper deserves consideration for publication.

The main point of the paper is that science advances through surprise, yet there were little to no examples of "surprise" and novelty. Some specific examples of surprise would help motivate and anchor the approach proposed in the paper. The authors do include the sanity check with the Faculty1000 papers, and I appreciate this inclusion but I still felt that this fell short of (1) providing some specific examples of surprise to anchor these models on and (2) some specific examples from their analysis that align with what others consider surprising. The vast majority of the paper is spent in hypergraph space, which is still valuable, but it would be helpful for an already interesting paper to show qualitatively the surprise that is referred to throughout the paper.

I am concerned that "surprise" of unique ontological combinations are simply the unique combinations that made it through peer review. It is not that unique combinations are what advance science. There are likely all sorts of "surprising" combinations but many of them are crackpot ideas. The few that get through are therefore more likely to be successful since they were novel and they made it through the peer review filter. I don't expect the authors to have to test this since pre-peer-review papers are nearly impossible to get, but I am concerned that this "surprise" is being slightly oversold.

How do others build upon this approach? What other questions can be asked? Does this change how we do science or just how we understand it? The authors conclude in their abstract that the work in this paper provides new tools "to evaluate how scientific institutions including awards, education and peer review facilitate advance." I did not find this in the paper. How would these tools allow for this?

The authors say that "where we show that the vast majority of new scientific and technological predictions can be predicted." Also, "accurate prediction of subsequent scientific discoveries and inventions." This doesn't accurately reflect what was done in this paper and what was not done. It does do a good job predicting ontological combinations, but it doesn't evaluate actual paper content. The claims of the paper title and abstract seem to exceed what is actually done in the paper. I think this needs to be toned down. It won't eliminate the novelty that does exist in this paper (e.g., the hyper graph representation and the various forms of representing surprise and discovery and ways to evaluate these measures). It just makes clear what was actually done. The authors have not created a method that predicts tomorrow's surprising discoveries.

One of the key sentences to understanding this paper is the following sentence: "If predictions are sufficiently accurate, those that cannot be forecast will surprise the community of scientists and vinegars who themselves seek priority by anticipating the future." What do the authors mean by

sufficient predictions? Do the authors mean predictions of future discoveries and inventions? And if so, are the authors implying that they have a method that succinctly predicts future discoveries? The authors are able to predict ontological combinations term but not necessarily future discoveries and inventions. Are these predictions sufficient? If so, a little more nuance here is needed.

Methodological question: The authors measure the likelihood of hypergraph G looking at every possible combination. How do these likelihoods vary across the various data sets investigated? The paper refers to the methods, but I couldn't seem to find this. I ask because I am curious how the variance of each affects the likelihood of surprise in each of the different hypergraphs. Also, the authors report that they correctly distinguished combinations that ended up as publications more than 95% of time. I am curious how much this changes for years in which there is an increase in the MESH terms or USPC codes.

I think the paper could do a better job differentiating content and context novelty, especially since it is one of the core concepts in this paper. There is sufficient information in the paper provided to figure out what the differences are, but it took more effort than is needed. The current explanation leaves room for multiple interpretations.

Related to the comment above, the authors claim that novel combinations are 4 times more likely to be a hit paper than random, but this does not account for the many "novel" combinations that make it no where (e.g., get filtered through peer review or fall to lower tiered journals not in the APS or indexed by MEDLINE. I don't expect the authors to measure this. I just note this to make sure that this 4 times results isn't oversold too much.

Sentence fix: "our work formalizes the concept of a 'knowledge expedition', where scientists from one area travel to distant another..."

In Figure 3b, is the entropy of attention affected by the increase in available terms over time?

We thank the reviewers for their insightful questions, comments and suggestions. Responding to these and expanding our analysis, sensitivity evaluations, and broadened framing has dramatically improved the clarity of presentation and demonstrated robustness of our analysis.

Reviewer #1 (Remarks to the Author):

This manuscript studies the relationships between "surprise" of scientific or technological discoveries and their success using a generative hypergraph model. The hypergraph is defined on the space of ontologies or classification systems (e.g., MeSH terms or patent subclass code) or venues (journals). The generative model defines, given the parameters (embedding of terms and the scalar reflecting the degree), a propensity function of possible hyperedges with a Poisson model (i.e., number of papers or patents with the given set of terms). The paper shows that the model can learn useful embedding from the past that can distinguish realized hyperedges from unrealized ones in the next period. The propensity function is computed as a product of "complementarity" and "availability". Each node is represented as a normalized latent vector. The complementarity is defined by examining whether all nodes in the combination are expressed in the same dimensions; the "cognitive availability", which roughly prescribes the degree of nodes, is modeled with a scalar parameter for each node (and their product for a hyperedge). Finally, the propensity of a hyperedge is computed as a product of these two factors. Two separate hypergraphs are considered to capture the "content" and "context". These graphs are separately defined for each dataset (e.g., MeSH terms vs. journals; PACS codes vs. journals; and subclass codes vs. class codes).

We are grateful that the reviewer so clearly understands our research design and intention.

Finally, novelty is defined as the surprisal of its complementarity. With the novelty defined for the content and context, the authors showed that the novelty and citation are positively correlated. Namely, as the citation increases, the mean novelty also tends to increase (although content and context novelty are not strongly correlated).

This is absolutely correct and one of our key findings. Prior work has elided these two variables, using what we call context (journals/conferences) to measure content, but we demonstrate how and why these represent significant, separate signals for the assessment of novelty and the prediction of outsized impact.

The probability of hits also increases with novelty. Finally, it was shown that the novelty measures defined for a team, career, and "expedition" capture some associations with the hit probability.

We are pleased to see that this sociological finding was also clear—that novel careers, teams and especially expeditions were substantially correlated with outsized (hit) novelty.

This paper proposes an intriguing, very clever hypergraph embedding model to capture knowledge space and its exploration. The combination of the hypergraph-based operationalization of knowledge products and the "complementary" formula that specifically captures the multi-dimensional combinatorial property seems like an innovative way to address the challenges of operationalizing novelty, moving beyond the existing combinatorial definitions of novelty. The model is validated by using a temporal link prediction task (using a model trained on the past to predict future hyperedges) and the result seems to indicate that the model indeed learns useful embeddings for predicting knowledge products. The results on the association between novelty and success are also fascinating. Therefore, I believe that this is an important contribution to the science of science.

Thank you; we are delighted that the Reviewer understood and appreciated our approach to modeling the hypergraph of knowledge products.

At the same time, I am unsure about several results and their presentations and I think it is important to clarify them before recommending them for its publication.

- I'm guessing that the vast majority of new hyperedges overlap with the existing hyperedges from the prior years. Is this true? One can think of a simple baseline where we predict "Yes" for previously seen hyperedges and "No" for previously unseen hyperedges. How well can this baseline perform in the evaluation task? I think the performance of this baseline would provide a useful context to the performance of the model.

We appreciate the question and the excellent associated suggestion. For each year, we now calculate the number of repeated hyperedges from prior years and include the result in the Supplementary Information (also attached below). We find this number to be lower than expected. For both the content and context hypergraphs of MEDLINE, less than 15% of hyperedges from any given year overlap with hyperedges from prior

years. When given a random hyperedge and a random non-hyperedge combination, there is at most a 15% chance that the hyperedge is repeated from previous years. If we predict “Yes” for all repeated edges and “No” for previously unseen edges, then at most 15% of the time we will rank a random hyperedge higher than a random non-hyperedge combination. This translates to an AUC of approximately 0.15, which is much, much lower than the performance of the current model.

- Across all figures, whenever possible, I would suggest showing the uncertainty of the estimations. Without uncertainty, it is difficult to judge whether the pattern is real or not.

We thank the Reviewer for this request and have since added confidence/credible intervals or standard errors to the plots whenever possible and also note that uncertainty in the figure captions (e.g., Figure 2(a), (b) and (c), Figure 3(a), Figure 4(a)).

- In Fig. 2, (a) and (b) show only the average value per decile. I think there's a missed opportunity to reveal more information -- how they are related in more detail. What's the uncertainty here? How is the novelty distributed for each citation decile?

We thank the Reviewer for this superb suggestion. We originally omitted the distributions in the interest of space, but now we have added distributions of novelty scores for each citation decile for MEDLINE in the SI (also attached below). Within each citation decile, we fit Gaussian kernels to the novelty scores and plot the resulting density function. We can see that the mass of the novelty distributions gradually shift towards the right (higher novelty) as we move from lower to higher citation deciles. This shift is clear for both content and context novelty—but more striking and dramatic for context. The density plots also suggest that uncertainty is relatively low for either lowest or highest citation deciles and higher for middle deciles. This inverted “U” shape of uncertainty is unsurprising: In lowest (and highest) citation deciles, the novelty scores

are also low (and high) for most papers; in the middle deciles, the papers are almost “uniformly” distributed across the novelty spectrum.

What happens if you flip the x and y-axis? Given their novelty, do prize winners tend to have fewer citations?

As shown in Figure 2 (a) and (b) and briefly mentioned in the text, most of the award-winning papers are in the top citation decile. Therefore, they tend to have (far) more citations than average in general. To confirm that they also have more citations for any given novelty score, we have followed the suggestion and flipped the x and y axes in Figure 2. The resulting plots are attached below. Prize winners have higher-than-average citations across novelty deciles. This observation agrees with the notion that citation is a complex outcome, and being a prize winner has a substantial impact on citation count. However, we can see that novelty is still significantly correlated with citation, even for most prize winners—a trend heightened for context novelty.

Have you examined the whole distribution (say, a heatmap of the citations similar to (d) or a 2D version of it)? Wouldn't this show useful details about the distribution of novelty and citations?

We thank the Reviewer for this interesting idea. We focused on the direct correlation between novelty and citations, but not their distributional relationships. In the revision, we examine the joint distribution of novelty and citation across all MEDLINE papers and added heatmaps (similar to Figure 2d) to the SI (also attached below). The distributions are roughly bi-modal, with one mode around 0 citations and 0 novelty, and the other near maximal citation and maximal novelty. Most papers are distributed along the citation~novelty line. Besides, the heatmap for content novelty is “noisier” than for

context novelty, agreeing with our finding that context novelty has a stronger association with citation.

- In Fig. 2, (c) and (d) shows many data points as dots, but I could not find what they are. I'm guessing that each dot may represent a bootstrapped sample but the details on these points couldn't be found.

We thank the Reviewer for pointing out this failure to communicate Figure 2 details. Each dot represents a (novelty percentile, hit probability) data point from a certain year. We plotted our results in this way because our task is to predict combinations at year t based on all history before t . Accordingly, each data point in the plot is about a specific year. We didn't aggregate over all years but plotted them separately to show variation

over time (or rather, the robust pattern over time identified by our model). We have clarified this point in the figure caption.

- In Fig. 4, it is unclear how the "heat-cube" is colored. The authors should specify and justify the method to obtain the smoothed map.

We thank the Reviewer for noting this ambiguity. The heat cube is generated as a multi-sided heat map. We have added the details in Section "Methods/Probability of hit papers and patents" and referred to them in the figure legend. As a brief summary here, we first discretize the bounded space of (career novelty, team novelty, expedition novelty) into $20 \times 20 \times 20$ cells and then distribute all the papers into those cells based on their corresponding values for the three novelties. In each cell, we calculate the fraction of hit papers as the probability of being a hit. Analogous to Figure 2(d) where we regress probability of hit to content and context novelties, here we fit a polynomial function between the probability of hit and the career, team, and expedition novelties of each cell to obtain the average relationship:

$$P(\text{hit}) \sim \sum_{i,j,k=0}^1 \text{career novelty}^i \times \text{team novelty}^j \times \text{expedition novelty}^k$$

The resulting 3D heat cube comprises 6 2D heatmaps where color denotes the fitted probability of hit for any given values of (career novelty, team novelty, expedition novelty). These are then joined precisely on their matched boundaries to reveal the complex relationship between all three novelties. We now detail these construction details clearly in the supplement.

- The main manuscript says, "MEDLINE papers with maximal joint context and content surprise predict nearly 50% of the likelihood of being in the top 10% of citations". I was not sure what exactly this 50% number means. I'd suggest elaborating a bit more on the task setup and how exactly this number was calculated.

We regret this confusion and thank the Reviewer for pointing out our ambiguity. What the sentence intended to communicate is: On average, nearly 50% of the papers in the group of highest content and context novelties jointly are hit papers (i.e., papers with top 10% citations). This is supported by Figure 2D, where the y-value (probability of hit) is close to 0.5 at the top-right corner (which corresponds to the papers with the highest joint content and context novelties). We have rephrased this sentence for clarity and expanded on the discussion regarding Figure 2D.

- I think more elaboration on the concept and precise operationalization of 'content' and 'context' in the main text would improve the readability of the paper.

Although reading the method section clarified for me, I was confused about how they are operationalized. A clearer, more explicit presentation of 'content' and 'context' (hypergraphs) may be helpful for the readers.

We thank the Reviewer for this excellent expository suggestion. We have reorganized the writing and elaborated on the concept and operationalization of content and context in the main text. We also add a detailed explanation about content vs. context novelty below equation (2), when the definition of novelty is first introduced. Specifically, we have added:

*We further contrast scientific contents and contexts to refine our expectations about normal scientific and technological developments. By **content** we refer to the substance of the papers and patents including phenomena, concepts, and methods. We operationalize these using community-curated ontologies—Medical Subject Heading (MeSH) terms for MEDLINE papers, Physics and Astronomy Classification Scheme (PACS) codes for APS papers, and United States Patent Classification (USPC) codes for patents, which index content for discovery within the field. By **context** we refer to scientific or technological fields that often bundle, index and publish research through editorial organization and disciplinary unity. We operationalize these as journals and conferences referenced within a paper (or technology classes referenced by a patent), which reflect the fields from which ideas and components were sourced. Then for each dataset in each year we build a hypergraph of contents where each node corresponds to an ontological term and each hyperedge to a paper or patent that inscribes a combination of terms. Meanwhile, for each dataset in each year we separately build a hypergraph of contexts where each node corresponds to a journal or conference (or major technological area for patents) and each hyperedge to a paper or patent that cited those contexts within its references.*

The differentiation between contents and contexts allows a more precise characterization of a novel discovery or invention. A new combination of contents may surprise because it has never succeeded before, despite having been considered and attempted; A new discovery or invention that cuts across divergent contexts may surprise not because it has never been attempted, but because it has never been imagined. The separate consideration of contents and contexts also allows us to contrast scientific discovery with technological search: Fields and their boundaries are clear and ever-present for scientists at all phases of scientific production, publishing and promotion, but largely invisible for technological invention and its certification in legally protected patents.

[Below Eq. (2)] Recalling that we model contents and contexts separately - a paper is simultaneously a combination of contents (e.g., MeSH terms) and a combination of contexts (e.g., journals), we separately measure the content and context novelty of a paper, which correspond to the novelty of its content and context combinations, respectively.

- I would encourage authors to consider rephrasing the claim that "the vast majority of new scientific and technological configurations can be predicted". Yes, it was shown that the model can distinguish the combinations of MeSH terms that happen from random ones, but I think the term "configuration" can mean many different aspects of knowledge products and thus the statement can be easily misinterpreted. Also, depending on the performance of the super simple baseline, this fact may potentially be much less surprising than it looks.

We agree with this concern and regret the unintended overstatement. By configurations, we only meant combinations of contents and contexts as defined in this work. We have rephrased that claim to be precise and clear regarding its scope: *the vast majority of new combinations of scientific and technological terms can be predicted*. Also, as shown in the first point above, the simple baseline model (predicting "yes" for all repeated hyperedges) would do much more poorly than our hypergraph embedding model. This comparison suggests that although future combinations might not be impossible to predict, neither is it trivial to do so.

- The name "block model" is a misnomer. In block models, the probability of an edge (propensity) only depends on each node's block membership and that's why they are called block models. Here, the proposed model does not have 'blocks' that dictates the propensity. It is rather just a hypergraph embedding model, similar to the geometric embedding models (e.g., hyperbolic space embedding models, word embedding models, etc.). I think the current name would confuse those who are familiar with block models. Therefore, I'd strongly encourage the authors to change the name to avoid unnecessary confusion.

We regret that this allusion was disorienting to the Reviewer. We named our model after the mixed-membership stochastic block model, where a node can belong to multiple `blocks` with fractional membership in each block. Such models have become important in recent years in the physical, biological and social sciences (Decelle et al. 2011; Abbe, Bandeira, and Hall 2016; Matias and Miele 2015; Celisse, Daudin, and Pierre 2012; Abbe and Sandon 2015; Abbe 2017; Lei and Rinaldo 2015; Mossel, Neeman, and Sly 2012; Latouche, Birmelé, and Ambroise 2011; Rachel Wang and Bickel 2017; Aicher, Jacobs, and Clauset 2013; Stanley et al. 2016). The boundary between such a `block`

model and an embedding model is blurred in this case, but we agree that the name may cause more confusion than clarity. We have changed the name to hypergraph embedding model, which indeed better reflects the fundamental idea of the model: embed the content and context nodes in a latent scientific knowledge space. We thank the author for this recommendation.

The method uses negative sampling, which is a biased and simplified version of the Noise-Contrastive Estimation (Gutmann & Hyvärinen, 2010). Although this may not be a huge issue because word2vec and other models with negative sampling work fine, the authors may want to discuss the fact that the negative sampling is a biased estimator and there may be certain consequences due to this bias.

Thank you for this thoughtful note. Negative sampling effectively changes the objective function we optimize. The original objective function can be rewritten as

$$\log P(G|\theta, R) = \sum_{h \in E} \log P(x_h|\theta, R) + \sum_{h \in \bar{E}} \log P(x_h|\theta, R)$$

where E is the set of hyperedges and \bar{E} is the set of non-hyperedge combinations. Negative sampling turns the second term from the objective function into a random function

$$X = \sum_{h \sim P(h|\bar{E})} \log P(x_h|\theta, R)$$

where h is randomly drawn from \bar{E} . This random function is a biased estimator of the original objective in negative sampling. To illustrate the bias, assuming the simplest case of stochastic gradient descent where we draw 1 negative sample for each positive sample, the expectation of X then becomes

$$\sum_h \log P(x_h|\theta, R)P(h) = \frac{1}{|\bar{E}|} \sum \log P(x_h|\theta, R)$$

where $|\bar{E}|$ is the size of \bar{E} and $P(h) = 1/|\bar{E}|$ because h is a draw from \bar{E} uniformly at random. The result for more complex stochastic gradient descent algorithms can be derived similarly with more complicated factors in front of the second term. In general, the effect from the second term is down-weighted due to negative sampling. Although statistically this estimator is biased, it works well in practice for word embeddings [Mikolov et al. 2013], question answering [Bordes et al. 2014], and many other applications, improving on predictions over noise contrastive sampling because it reduces variance (Mikolov et al. 2013). We also found this approach worked well in our model as seen in the high predictive power despite bias. Nonetheless, we acknowledge that the estimation could be further improved by considering more advanced sampling schemes [Gutmann & Hyvärinen, 2010], but we defer this to future work. There is also

research on the comparison between different sampling methods [Wunsch et al. 2009, Saeidi et al. 2017], but it is beyond the scope of this paper to precisely quantify biases due to negative sampling within our approach based on our focus on prediction accuracy rather than parameter estimation. We note this caveat in the revision and have incorporated this discussion into the Methods section.

- Not much detail was provided regarding the details of model training and thus I don't think it is possible to replicate the results just based on the paper. The code may provide all necessary details, but the GitHub address is not disclosed in the paper. Could you specify hyperparameters? For instance, what are the hyperparameters for SGD? What are the largest hyperedge size for each dataset? What's the hyperparameters for the negative sampling?

These are all excellent questions. We have provided the details of training in the SI. The GitHub repo is also included in the revised Code Availability section to assist with replication. Here is a brief summary.

For stochastic gradient descent

- Learning rate is set to $1/(100+\text{number of epochs})$ for all datasets.
- Batch size is set to 1000 for the MeSH dataset and 2000 for all the others.
- Number of negative samples is set to 1 per positive sample.

For other training hyperparameters

- Number of training epochs = 50 for all hypergraphs from the MEDLINE dataset and the US Patent dataset
- Number of training epochs = 25 for context hypergraphs from APS
- Number of training epochs = 100 for content hypergraphs from APS
- Maximum hyperedge size = 47 for MEDLINE MeSH, 711 for MEDLINE journals, 15 for APS PACS, 125 for APS journals, 264 for Patent subclass dataset, and 207 for Patent class dataset.

- Right after Eq. 2 says, "We note that our deep learning-based transformer predictions of content and context ...". But I believe that the manuscript does not talk about the transformer model at all, but it is only mentioned in the SI. I think this paragraph is not necessary in the main manuscript.

We have removed this paragraph from the main text. We were just hoping to clarify that these methods work for contemporary contextual embedding methods as well.

- In Figure 1 (b), I think the green, not the blue, is the most novel article.

Good catch. Yes, green is the most novel. This is a typo and now fixed in revision.

Reviewer #2 (Remarks to the Author):

I am pleased to submit a review of this very interesting paper, "Science and Technology Advance through Surprise, driven by Expeditions of Outsiders." The paper takes computation approach to search for patterns in large amounts of scientific articles and patents using existing ontologies for scientific fields and technology areas. The approach seeks to develop a model of abduction following C.S. Pierce.

We are grateful that the Reviewer found our purpose clear and our paper interesting.

(My copy did not have page numbers so I numbered the pages starting with the abstract on page 1.)

The paper's findings depend on the success of a model created for experimentation with a large corpus of scientific output including journals, articles, and patents. It is difficult for this reviewer to assess the propriety of the model. It would be difficult to draw from this paper what other possibilities for experimentation might have been possible. Is this the best model for assessing patterns - it appears to be an emergent model, although there is reference to higher order structuring (p.4). It is not clear at what level this paper is operating if one assumes a hierarchical model.

We note that we did not borrow or repurpose this model from another application. We designed and implemented it specifically for its purpose in this article—to predict the combinations of contents and contexts across science and technology. In the paper, we show that the model strongly outperforms other models in the literature with similar purposes (see “Surveying predictive models of hit papers” in SI), but we avoid claiming that it is the best possible model for this task, despite not being aware of a better one.

In most social science fields, scholars avoid using the term "predict" because of the existence of non-linearity and multivariate conditions, thus it concerns this reviewer to see the term 'predict' used a number of times in the paper. It seems as if the model is set up to search for what it finds, although I may not understand it fully.

We thank the Reviewer for raising this concern and regret the confusion raised over our use of prediction in this paper. The rise of powerful, predictive machine learning models has led to a growing interest in the task of prediction across the social and policy sciences (Kleinberg et al. 2015; Molina and Garip 2019; Ettensperger 2020; L. Zeng

1999; Selin and VanDeveer 2007). Forecasting and prediction have long been goals of specific social science areas including finance and election research, but the task has diffused more broadly with increasing prediction success, as driven by the much wider range of multi-modal data than ever before (e.g., contents and contexts as we use here). Specifically, we are hindcasting (predicting the past from the more distant past) rather than forecasting (predicting the future from the present). We note that our models are neither causal nor descriptive, which are important social science goals, but not relevant to the prediction portion of our analysis. In that portion, we seek specifically to predict future science in order to identify the most *unpredictable* or surprising papers and patents. As a result of this comment, we now do a better job of identifying our research goals and the importance of prediction for our specific task of measuring surprise.

The idea of a complex hypergraph of scientific advance is intriguing and appears to be an interesting approach, but, in the paper itself, the methodology section only briefly describes the process. As this reviewer reads it, the method seeks clusters of terms - is that correct?

Yes; that is correct. More precisely, the method seeks to embed the terms in a latent space (i.e., assign numeric vector representations) so terms that co-appear in papers more frequently will be closer in the space. We have expanded our Methods sections, and also added the complete code for replication for complete clarity.

The clusters are derived from MeSH terms?

The content “clusters” are of MeSH terms for the biological sciences; PACS codes for the physical sciences (a MeSH equivalent for the physical sciences); and USPC codes for patented inventions. The context clusters are of citation sources (e.g., *American Economic Review*, *Cell*, *Nature*, etc.) We have added more details about our data in the Data section in Methods.

The terms are placed into datasets by year to build a hypergraph for that year, although I am not totally clear on this method. Reading the more in-depth method appendix still leaves it unclear as to what variables are being examined, and how many different experiments were run. Can you please clarify the variables and experiments?

We have markedly increased both the Methods section in the paper, and also the Supplementary Information to clarify exactly what we did to perform our analysis. We have also linked to our complete code base, which can be used for precise replication.

In our analysis, we calculated embeddings (numeric vector representation) of keywords (and, separately, referenced source journals) from research papers and patents. Different from traditional regression analysis where independent and dependent variables are defined explicitly, here we only have the dependent variable, which is the number of times a combination of terms appears in papers (including zero times). Roughly speaking, the “independent variables” are discovered through creation of the embeddings (i.e., the latent numeric vectors) of terms or journals in an article “combination”. Nevertheless, the embeddings are not given *a priori* but estimated from data. Furthermore, not every combination has the same number of terms or the same number of “independent variables” as commonly seen in a regression analysis. Hence, we recommend not comparing our model to a typical regression model but rather a machine learning one. In short, we predict future combinations of terms (and journals) based only on an embedding of prior combinations. This allows us to identify papers and patents that are unpredictable or surprising, and we show that this corresponds with assessments of novelty from the scientific community.

As for the number of experiments, take MEDLINE data as an example: For each year t , we fit the model to all papers published before and in year t focusing on MeSH terms (contents), and separately, we fit the model to all papers published before and in year t focusing on cited journals (contexts). Therefore, we estimated T^2 models for the MEDLINE data where T is the total number of years covered by the analysis. The same procedure is followed for the APS and USPTO datasets. Specifically, we run 68 experiments with MEDLINE data, 40 experiments with APS data, and 46 experiments with USPTO data.

The reference in page 1 to serendipity of science is apt, but there have been many different papers and experiments delving into the dynamic structure of science beyond Merton. These works do not seek to predict what will happen in the future -- that is not the goal of any known research in sociology of science.

In the burgeoning “science of science” literature (Fortunato et al. 2018)—including work from sociology and economics, but also computer and data science, physics, ecology, and more—there is a growing body of work that seeks to predict the number of future citations by researchers, institutions and journals (D. Wang, Song, and Barabási 2013; Li and Agha 2015); the time at which those citations are received over the life course of scientists (Liu et al. 2018; Sinatra et al. 2016); and factors that influence impact predictability (Rigby 2013). While most of these do not describe themselves as the “sociology of science”, they nevertheless incorporate social factors in making increasingly successful predictions of impact.

However, the work of Leydesdorff, in particular, deals with the complex dynamics of communications structures such as self-organizing criticality of discovery and Shannon uncertainty in communication structures - and either of these concepts would be good to discuss in light of the approach suggested in this paper.

We thank the Reviewer for this reference and now discuss the work of Leydesdorff and colleagues regarding the emergent, complex system of discovery in science (Loet Leydesdorff 2010; L. Leydesdorff and Van Den Besselaar 1997; Etzkowitz and Leydesdorff 2000; Loet Leydesdorff 2000)

The contrast with patent data is interesting, but it might be better to have a separate paper for the patent data, and only refer to it briefly in this paper. The discussion does not apply directly to the focus on surprise in scientific advances. As the authors note on page 7, expectations for patents is quite different than for papers. Moreover, it is unclear how the patent discussion supports the initial goal of demonstrating abduction.

We appreciate the suggestion of the Reviewer regarding a deeper dive into patent dynamics in another paper and agree that it will be an interesting topic to explore next. We regret that we did not better communicate the three important roles that patents play in the current paper. First, the patent data demonstrates that our hypergraph and surprise models are general and flexible - they not only capture scientific discoveries but also technological inventions. Second, the contrast of patents with science highlights a few distinguishing patterns about science. For example, we find that scientists cite contexts similar to publishing venues more intensively than contexts that are distant (Figure 3a). It would be hard to assess whether this pattern is substantial without comparing it to the patent space where inventors cite close or distant sources with equal likelihood. Finally, the lack of disciplinary boundaries in patents helps us understand how the surprise of context combinations distinguishes novel scientific discoveries. As a result of your comment, we now do a better job in the paper of articulating how our analysis on patents provides an important contrast in understanding how science advances. We look forward to follow-up studies (from us or others) with patents being the focus that more thoroughly investigate how technology advances.

On page 3, the text reads: "The model implies..." this may be a small thing, but a model cannot imply something.

We thank the Reviewer for catching this unintended ambiguity. We have changed the language for clarity to "The successful prediction of future combinations implies ...".

The next sentence says that local search agrees with previous findings published by one of the authors. In fact, the concept of local search in scientific communities has a long history in sociology of science, so it might be better to cite other work preceding this paper rather than the author's own work.

We thank the Reviewer for this recommendation. We now add referenced work from the science studies, sociology and complex systems traditions (Callon, Rip, and Law 1986; Guimerà et al. 2002; Stuart and Podolny 2007; Jia, Wang, and Szymanski 2017)

I very much like the finding on page 5-6 on content and context differences in their relationship to outcomes. The paper does not set this up as a hypothesis - seeking only abduction - but the relationship between content, context and outcomes is an interesting insight. Can you say what the significance measures are for this finding? How does it relate back to the initial hypothesis?

We thank the Reviewer for this perceptive question. We have added 95% confidence intervals around the line plots in Figures 2a, b, and c (which describe robust associations between novelty and citation impact or awards). Confidence bounds suggest that the positive associations between content/context novelty and citation or hit probability are significant and substantial. Specifically, we test the statistical significance of the Pearson correlation between content/context novelty and hit probability in Figure 2c, and the results are $\rho = 0.90$, $p \approx 0$ for content novelty against hit probability and $\rho = 0.87$, $p \approx 0$ for context novelty. Similarly, for mean content novelty against citation quantile (Figure 1a) $\rho = 0.98$, $p \approx 0$ and for mean context novelty against citation quantile (Figure 1b) $\rho = 0.99$, $p \approx 0$. Last but not least, the confidence bounds in Figure 2c also suggest that the effects of content and context are significantly different.

To the Reviewer's question regarding our initial hypothesis that "path-breaking science begins as expectations are disrupted by surprising findings, which then simulate scientists to forge new theories to make the surprising unsurprising," a logical consequence of this hypothesis is that the surprise of a discovery (if properly measured) would be able to "predict" its impact and influence on science and scientists. Here using citation and awards as imperfect measures of impact, we show that surprise, as measured by our model, is the best available predictor of outsized impact. The finding is consistent with our initial hypothesis. We separated content and context because prior studies have used context as a proxy for content in estimating novelty (Uzzi et al. 2013), but we show that surprising combinations of content and/or context have largely separable and independent influences on characterizing expectations and perceptions of novelty. For this reason, we argued that an improved measure of surprise would take

them both into account. In this current draft, we increase discussion linking our content and context measurements with surprise; making it more clear why the distinction is important for our purposes in this paper.

The discussion of teams and expeditions is also very informative. The findings make some contributions to the science of teams. Here, I wonder what 'career novelty' and 'team novelty' consists of, since that is not defined, page 6. It has been shown that the work of teams has higher impact over time than the work of sole authors. The findings here seem to support that work. Perhaps a reference to that earlier work would be helpful here.

We regret this omission and thank the Reviewer for pointing it out. In the previous version, we only briefly introduced the career, team and expedition novelties on page 6 due to the space constraints. Now we have added formal definitions of the three novelties in Methods and refer to them clearly on page 6. We have also cited more work related to novel, exploratory careers (Liu et al. 2021; A. Zeng et al. 2019) and novel teams (de Vaan, Vedres, and Stark 2015; Uzzi and Spiro 2005).

On page 7, the authors refer to collective abduction, which seems like an intriguing idea. It might be worth considering defining this term fully and explaining how the operation shows this feature. Similarly with the idea of 'collective attention.' Here, the paper also discusses exposure to foreign theories that enable surprising discoveries. Again, it would be good to have more information here. Several background papers might be worth consulting about surprise, including "Being Surprised and Surprising Ourselves," by Dragos Simandan (2018), and "Tools for Thought" by H. Rheingold (1985).

These are excellent suggestions and have inspired us well beyond the scope of this project. They moved us to write an entire paper on collective abduction in science (Duede and Evans 2021), which we both reference here, but also add some additional parsimonious discussion in the paper for clarification (here is the online reference: <https://arxiv.org/abs/2111.13251>; the paper is under first review *Social Studies of Science*). In short, abduction requires both insiders to recognize the surprise, and outsiders to bring their alien skills to make the surprising unsurprising—to solve the problem. The concept of collective attention is more common, but typically referenced in the social media literature and operationalized as shared mentions (Wu and Huberman 2007; Lehmann et al. 2012; Mocanu et al. 2015). Here we proxy it with content—more people referencing a concept suggests that more people are attending to it. We also add the very interesting references the Reviewer proposed regarding the perception of surprise (Simandan 2020; Rheingold 1985).

Also on page 7, the authors refer to cross-disciplinarity, interdisciplinarity, and multi-disciplinarity. Here, no doubt, the authors are aware of a deep literature on these subjects. Measures have been developed that could also be applied to assess changes in the knowledge base. A discussion of these methods as they are similar to or different from the method proposed here would be enlightening.

We thank the Reviewer for raising this excellent point. In our initial draft, we considered ourselves constrained by perceived space limitations, but now have created an expanded discussion of alternative measures of X-disciplinarity, and how our findings articulate with findings from this burgeoning literature, recently reviewed in (Q. Wang and Schneider 2020). This includes prominent measures that approach interdisciplinary from the perspective of combinations of journal keywords from the Web of Science (Morillo, Bordons, and Gomez 2001; Morillo, Bordons, and Gómez 2003; Porter et al. 2007; Leahey, Beckman, and Stanko 2017) Porter (Morillo, Bordons, and Gomez 2001; Morillo, Bordons, and Gómez 2003; Porter et al. 2007; Leahey, Beckman, and Stanko 2017) and diversity and inequality indices (e.g., the Simpson Index, Shannon Entropy, and the GINI coefficient) assessed over these categories (Loet Leydesdorff and Rafols 2011; Zhang, Rousseau, and Glänzel 2016). These relate to our work, but are much less granular (ours involves article keywords; their journal keywords). Others account for the similarity or difference between research fields combined (Porter et al. 2007; Porter and Rafols 2009; J. Wang, Thijs, and Glänzel 2015). Our measure does this automatically by accounting for the combinations of keywords required to characterize current literature and predict future research. Still others are built atop measures of centrality in networks of publication (Loet Leydesdorff 2007; Rafols et al. 2012). By measuring the inner product of term vectors, our measure also approaches a continuous measure of centrality within the embedding space of contents and contexts. In short, our measure captures desiderata from each of the major categories of novelty measurement—diversity of combination, accounting for differences between components, and which represents centrality in the continuous embedded manifold. We now discuss this in the Methods section.

On page 8, the authors suggest that the method could be applied to institutions to help facilitate progress and also suggest that the method could help foster advances. It may be premature to suggest a normative application of this method. The model offers an intriguing way to study the structure of scientific advance, but it is early days in testing it; its possible applications lay in the future. Institutions generally do not want to have metrics of 'progress' put on their creative work.

We thank the Reviewer for pointing out this overstatement. We note that we are not proposing an artificial intelligence to score (and determine) everything in science and technology. Rather, we hope that our novelty measure can provide a complement to other conventional measures such as impact factor and h-index to offer a more holistic view of scientific processes that value novelty as well as popularity. Moreover, researchers can apply our methods to specific research problems such as studies of the peer review system or funding mechanisms to advance science in other ways. Nonetheless, we note that agencies such as NIH and NSF are looking for metrics to rank intellectual products given the increasing workload they are facing. Those being said, we completely agree that it is too early to suggest a normative application of our method, which is not what we intended to convey. We have revised that part in the paper to better reflect our more modest scientific intention.

Similarly, also on page 8, the finding on prizes going to conservative advances, and that people cite familiar sources, have both been shown in the sociology of science in several different contexts, so it might be good to cite these works.

We thank the reviewer for mentioning this omission. We have now included references to the following work on the conservatism associated with: 1) prizes in science (Zuckerman 1978; Ma and Uzzi 2018; Szell, Ma, and Sinatra 2018); and 2) citations more broadly (Bornmann and Daniel 2008; Jannot et al. 2013; Nieminen et al. 2007; Brysbaert and Smyth 2011; Case and Higgins 2000).

The idea of knowledge expeditions (while not termed in this way exactly) also supports other work that has been done in the past. It is unclear why the authors see their findings as distinct from interdisciplinarity since no effort has been made in the paper to distinguish findings here from others. However, it has been noted that researchers search for new ideas and that creative researchers put together work across structural holes (Burt, 2002). It has also been shown that Nobel Prize winners span structural holes (Wagner et al., 2015).

We thank the Reviewer for highlighting this omission. We reference these excellent works on bridging structural holes in science and problem solving (Burt 2004, 2021; Wagner et al. 2015), and reveal their alignment with our findings on knowledge expeditions.

Small item, but footnote 36 does not appear in the references.

We have corrected this error. Reference 36 is now included in References as

36. Zivin, J. G., Azoulay, P. & Fons-Rosen, C. Does Science Advance One Funeral at a Time? *Am. Econ. Rev.* 109, 2889–2920 (2019).

It was disappointing to get to the end of the paper and not find a discussion of how the model upholds or refutes the concept of abduction. This appeared at the beginning to be the purpose of the article, yet, there is no further discussion about this hypothesis in light of the workings of the model. The idea is intriguing, and it would be good to say more at the opening as to how the model will reveal abduction. The conclusion should say something about the hypothesis and the extent to which it reveals this dynamic.

We thank the Reviewer for noting this discrepancy and deeply regret not returning to our initial framing regarding abduction and surprise in the conclusion. We have remedied this omission by demonstrating how our model demonstrates that science does substantially attend to surprising or unpredictable combinations of contents and contexts.

Overall, there are many observations and findings in the paper that could fruitfully refer back to earlier theories, models, and concepts in the dynamics of scientific advance. Some of the results support earlier work and other points refute it. Scholarship would be greatly advanced by a discussion of how the model tests and challenges these earlier concepts. Moreover, many of the earlier works have been written by women, and thus it was disappointing to see so few women scholars cited in the references.

We deeply regret this omission and have sought to remedy it. We initially modeled this paper on the *Nature Communications* template, which requires a modest reference list, but we have expanded it to reference other theories, models and concepts associated with scientific advance. We regret that our references in the initial manuscript were disproportionately to men, We have added several additional citations to female researchers working in this area, and we welcome other specific suggestions for missing references.

Women represented in our current reference list include:

Elinore Barber

Christine M. Beckman

Katy Börner

Erin Leahey

T. L. Stanko

Stefanie Haustein

Hyejin Youn
Paula Stephan
Deborah Strumsky
Reinhilde Veugelers
Roberta Sinatra
Yuening Hu
Xia Hua
Eva Guinan
Niki Parmar
Harriet Zuckerman
Cassidy Sugimoto
Staša Mikojević
Heidi Williams
Helga Nowotny
Camille Limoges
Kelley Packalen
Kjersten Bunker Whittington
Ishani Aggarwal
Anita Williams Woolley
Danielle Li
Laurel Smith-Doerr

I look forward to seeing this work develop in the months and years ahead. Good luck with it.

We thank the Reviewer for these excellent, engaged comments and suggestions. We believe that responding to them has made a more engaging, comprehensive and inclusive piece.

Reviewer #3 (Remarks to the Author):

This paper examines the role of surprise in advancing scientific knowledge. It combines a series of methodologies from machine learning and statistics (e.g., embeddings, hypergraphs, stochastic block models, attention-based neural networks) using community-curated ontologies to investigate whether various content/context combinations and whether they can be predicted. (I wonder if this paper's unique combination of various methodologies will increase its probability of success?) There is a substantial amount of work that has been invested in this paper with large-scale data and complex analysis tools. Overall, I find the paper a nice contribution to science studies. It will generate conversations about surprise, discovery and novelty in science and technology. My main concern is the methods are a bit oversold for what they say about surprise and discovery in science. If the authors can tone this down a bit and can address some of my other concerns, I think the paper deserves consideration for publication.

We are grateful that the Reviewer so clearly apprehended our approach and purpose in our analysis.

The main point of the paper is that science advances through surprise, yet there were little to no examples of “surprise” and novelty. Some specific examples of surprise would help motivate and anchor the approach proposed in the paper. The authors do include the sanity check with the Faculty1000 papers, and I appreciate this inclusion but I still felt that this fell short of (1) providing some specific examples of surprise to anchor these models on and (2) some specific examples from their analysis that align with what others consider surprising. The vast majority of the paper is spent in hypergraph space, which is still valuable, but it would be helpful for an already interesting paper to show qualitatively the surprise that is referred to throughout the paper.

This is an excellent suggestion. There are quite a few discoveries that motivate and align with our model, but we now include 3 immediately after defining the measurement:

In order to illustrate this measurement approach, consider exemplary discoveries our model identifies as novel. The work of Gryniewicz and colleagues (Gryniewicz, Poenie, and Tsien 1985) defines a novel family of chemical compounds including Fura-2, discovered to be highly fluorescent and bind to free calcium. This is a novel discovery—pulling diverse properties together that cross traditional field boundaries with a content novelty in the 99th percentile—receiving more than 16,000 citations to date. Research by Yanagisawa and colleagues (Yanagisawa et al. 1988) isolated endothelin,

one of the most potent vasoconstrictors at the time (1988), and employed several distinct and community spanning methods to examine its properties and mechanisms, finding its potential as a modulator of ion channels. This novel discovery is in the 95th percentile of content novelty and has received more than 14,000 citations since publication. Also, consider work by Altschul and others (Altschul et al. 1997) in which content and context novelties deviate. The work used a computer system to search protein and DNA databases and did not itself represent a novel discovery in the biomedical sciences but produced a tool with sizeable impact on the broad scientific community—it is among the 15 most cited papers of all time (Van Noorden, Maher, and Nuzzo 2014). In other words, its content is not the most novel from a biomedicine perspective, lying in only the 15th percentile of content novelty, but its influence spans the computational and biosciences fields and used techniques sourcing from those distinct communities, scoring in the 97th percentile of context novelty.

We have added those examples to the main manuscript.

I am concerned that “surprise” of unique ontological combinations are simply the unique combinations that made it through peer review. It is not that unique combinations are what advance science. There are likely all sorts of “surprising” combinations but many of them are crackpot ideas. The few that get through are therefore more likely to be successful since they were novel and they made it through the peer review filter. I don’t expect the authors to have to test this since pre-peer-review papers are nearly impossible to get, but I am concerned that this “surprise” is being slightly oversold.

We regret that it appeared we were overselling our surprise measure. We agree that one could come up with a fundamentally absurd, hopeless and doomed idea and our surprise/novelty measure cannot assess the validity of such ideas. However, conditional on a paper having survived through publication — having been filtered and validated through the adversarial peer review filter, our surprise measure is demonstrated to work well and reveal surprising successes that disproportionately reshape attention. It would be interesting to test our surprise measure on papers that did not make it through the peer review process but as the reviewer states, it is not possible to obtain such data and we will have to defer that to future, experimental studies once such data is available. In revision, we acknowledge this limitation clearly in the Discussion and throughout. The discussion text is below:

We note that we only evaluate our surprise measure on papers that pass the filter of peer review. We would like to test our model and measures on papers that did not make it through that process—presumably some containing bizarre or speculative association

culled in review. Unpublished historical papers in most fields remain inaccessible, and we defer their investigation for future research, but we expect that the success associated with surprise hinges largely on the collective skepticism that novel work must endure peer review (Merton 1942).

How do others build upon this approach? What other questions can be asked? Does this change how we do science or just how we understand it? The authors conclude in their abstract that the work in this paper provides new tools “to evaluate how scientific institutions including awards, education and peer review facilitate advance.” I did not find this in the paper. How would these tools allow for this?

We regret that we “skipped” on a discussion that would have allowed us to do some of the forward-looking integration you suggested. The short form of the article made it difficult to expand on all possible extensions and applications of our work, but we envision at least two promising directions along the lines of our approach.

Methodologically, we proved that higher-order information is important in modeling the space of knowledge through a relatively simple hypergraph model. Future studies can experiment with more advanced models such as deep neural networks and more signals such as textual data to model the advance of science. All those can be built upon the basic content and context hypergraph structures that we have developed here.

On the application side, our model provides novelty measures that are shown to be the most “accurate” in the literature to the best of our knowledge. Those measures can be used to quantify various intellectual products in science and technology such as papers, patents, proposals, etc., along with impact factor, h-index and other conventional measures to offer a more holistic view of the products. Researchers in science studies can apply our methods to specific research problems such as the role of novelty in peer review processes. In short, our model and measures provide a toolkit to evaluate the likelihood and surprise of any combination of objects. We have revised that sentence in the abstract to better reflect its meaning and expanded discussion of these possibilities in the abstract.

The authors say that “where we show that the vast majority of new scientific and technological predictions can be predicted.” Also, “accurate prediction of subsequent scientific discoveries and inventions.” This doesn’t accurately reflect what was done in this paper and what was not done. It does do a good job predicting ontological combinations, but it doesn’t evaluate actual paper content. The claims of the paper title and abstract seem to exceed what is actually done in

the paper. I think this needs to be toned down. It won't eliminate the novelty that does exist in this paper (e.g., the hyper graph representation and the various forms of representing surprise and discovery and ways to evaluate these measures). It just makes clear what was actually done. The authors have not created a method that predicts tomorrow's surprising discoveries.

We regret this bizarre overstatement. We have now made it clear throughout the manuscript that the target of prediction is combinations of keywords—and that even though our model is generative, our success measure only reflects our ability to distinguish actual papers from negative samples (random collections).

One of the key sentences to understanding this paper is the following sentence: “If predictions are sufficiently accurate, those that cannot be forecast will surprise the community of scientists and vinegars who themselves seek priority by anticipating the future.” What do the authors mean by sufficient predictions? Do the authors mean predictions of future discoveries and inventions? And if so, are the authors implying that they have a method that succinctly predicts future discoveries? The authors are able to predict ontological combinations term but not necessarily future discoveries and inventions. Are these predictions sufficient? If so, a little more nuance here is needed.

This is an excellent point. By that sentence we meant two things. First, we view our approach as a proof of concept that surprise and novelty can be quantified by accurately predicting normal science and technology in the future. If the predictions are poor or inaccurate, they will be unlikely to correspond with scientists' presumably better predictions about the future; and when they are violated (i.e., surprised), it is unlikely that we will correspond with scientists' experience of surprise. It is true that our method only predicts combinations of keywords (community-curated ontological terms), but even by doing so, we demonstrate that our measure performs better than competing measures at predicted outsized attention and even attributions of novelty (e.g., spontaneously scored “controversial hypothesis”.) If one can predict the complete, structured content of future discoveries and inventions, we expect their approach will provide an even more precise identification of novelty by calculating the surprise of the predictions. Secondly, by sufficient predictions, we mean accurate predictions, and our hypergraph embedding model does achieve high accuracy ($AUC \sim 0.9$) in predicting future combinations of keywords and journals against negative samples. Nevertheless, we agree with the reviewer that calling our predictions simply “discoveries” and “inventions” is misleading. We have reworded that paragraph and the one that follows to clarify what our hypothesis is and what is achieved by our method:

We frame surprise in science and technology as the violation of expectations about future advance. This demands that we predict the composition of future research with sufficient accuracy that what cannot be forecast will surprise the community of scientists and inventors who themselves compete for priority by anticipating the future (Merton 1957; Partha and David 1994). ... Here we model science and technology as a complex hypergraph drawn from an embedding of research components, as detailed below. (Tshitoyan et al. 2019) We separate paper components into scientific contents and contexts to refine our expectations about scientific and technological developments³⁰. By contents we refer to the substance of papers and patents such as concepts and methods, which we operationalize as keywords distilled within community-curated ontologies—Medical Subject Heading (MeSH) terms for MEDLINE papers, Physics and Astronomy Classification Scheme (PACS) codes for APS papers, and United States Patent Classification (USPC) codes for patents. Contexts refer to scientific or technological disciplines, operationalized as disciplinary journals and conferences referenced within a paper (or technology classes cited by a patent). For each dataset in each year we build a hypergraph of contents where each node corresponds to a content keyword and each hyperedge to a research paper or patent that combines all such keywords. Meanwhile, for each dataset in each year we separately build a hypergraph of contexts where each node corresponds to a journal or conference (or major technological area for patents) and each hyperedge to a paper or patent that references these disciplinary contexts as sources of inspiration and influence.

Methodological question: The authors measure the likelihood of hypergraph G looking at every possible combination. How do these likelihoods vary across the various data sets investigated? The paper refers to the methods, but I couldn't seem to find this. I ask because I am curious how the variance of each affects the likelihood of surprise in each of the different hypergraphs. Also, the authors report that they correctly distinguished combinations that ended up as publications more than 95% of time. I am curious how much this changes for years in which there is an increase in the MESH terms or USPC codes.

We thank the Reviewer for these excellent, engaged questions, and regret that our initial draft did not address them. The likelihood of a hypergraph is used as an objective to guide optimization of the model, but the likelihood itself does not carry much substantial meaning because it is affected by the training configurations such as batch size, negative sample size, etc. In other words, likelihoods of the hypergraphs from different datasets are not directly comparable. That is why we evaluate the models by their predictive power: their AUC scores in predicting hyperedges in the next year. AUC measures how often the model distinguishes a published combination from a random or

negatively sampled combination. We plot the AUC scores over time for the MeSH term hypergraph and the USPC hypergraph below (also added to SI: Extended Data Figure 13). As the reviewer requested, these plots also show the number of MeSH terms and USPC subclass codes over time. The number of MeSH terms has been increasing steadily over time but the model's AUC remains in a narrow band (between 0.94 and 1). The dip in AUC around 2001 could be numeric noise and the AUC is still above 0.94. The number of USPC codes and the model's AUC show more variation compared to the plot for MeSH terms, but the AUC scores are still above 0.84 and are not affected by the change in USPC codes.

I think the paper could do a better job differentiating content and context novelty, especially since it is one of the core concepts in this paper. There is sufficient information in the paper provided to figure out what the differences are, but it

took more effort than is needed. The current explanation leaves room for multiple interpretations.

We apologize that we overlooked this important issue in the draft. We have added a detailed explanation about content v.s. context novelty in setting up the distinction and below equation (2), when the definition of novelty is first introduced:

We separate paper components into scientific contents and contexts to refine our expectations about scientific and technological developments³⁰. By contents we refer to the substance of papers and patents such as concepts and methods, which we operationalize as keywords distilled within community-curated ontologies—Medical Subject Heading (MeSH) terms for MEDLINE papers, Physics and Astronomy Classification Scheme (PACS) codes for APS papers, and United States Patent Classification (USPC) codes for patents. Contexts refer to scientific or technological disciplines, operationalized as disciplinary journals and conferences referenced within a paper (or technology classes cited by a patent). For each dataset in each year we build a hypergraph of contents where each node corresponds to a content keyword and each hyperedge to a research paper or patent that combines all such keywords. Meanwhile, for each dataset in each year we separately build a hypergraph of contexts where each node corresponds to a journal or conference (or major technological area for patents) and each hyperedge to a paper or patent that references these disciplinary contexts as sources of inspiration and influence.

Distinguishing contents and contexts allows us to characterize the nature of a discovery or invention's novelty more precisely than before. A new combination of contents may surprise because it has never succeeded before, despite having been considered and attempted^{31,32}. A new discovery or invention that cuts across divergent contexts may surprise not only because it has never been attempted, but because it has never been imagined—representing a combination of ideas inaccessible within disciplinary conversation. The separate consideration of contents and contexts also allows us to contrast scientific discovery with technological search: Fields and their boundaries are clear and ever-present for scientists at all phases of scientific production, publishing and promotion, but largely invisible for technological invention and its certification in legally protected patents and marketed products.

...

As we model contents and contexts separately—a paper is simultaneously a combination of contents (e.g., keywords) and a combination of contexts (e.g., referenced journals), we also measure the content and context novelty of a paper separately corresponding to the novelty of its content combination and its context combination, respectively.

Related to the comment above, the authors claim that novel combinations are 4 times more likely to be a hit paper than random, but this does not account for the many “novel” combinations that make it no where (e.g., get filtered through peer review or fall to lower tiered journals not in the APS or indexed by MEDLINE. I don’t expect the authors to measure this. I just note this to make sure that this 4 times results isn’t oversold too much.

This is correct, and we clarify this point in both the setup and the discussion. As in the response to the comment above, we have clarified the scope of our novelty results in the revision. Here we also point out the limit of this “4 times results” in the corresponding paragraph in the revision. Our discussion now clearly states:

We note that we only evaluate our surprise measure on papers that pass the filter of peer review. We would like to test our model and measures on papers that did not make it through that process—presumably some contained bizarre or speculative association culled in review. Unpublished historical papers in most fields remain inaccessible, and we defer their investigation for future research, but we expect that the success associated with surprise hinges largely on the collective skepticism that novel work must endure peer review⁴⁹.

Sentence fix: “our work formalizes the concept of a ‘knowledge expedition’, where scientists from one area travel to distant another...”

Fixed.

In Figure 3b, is the entropy of attention affected by the increase in available terms over time?

This is an excellent question. The entropy of attention is not affected by length, as it is normalized by its information length (i.e., $\log(n)$ where n is the number of terms) so that it can be compared across years and datasets. We now note this in the caption of Figure 3b.

Abbe, Emmanuel. 2017. “Community Detection and Stochastic Block Models: Recent Developments.” *Journal of Machine Learning Research: JMLR* 18 (1): 6446–6531.
Abbe, Emmanuel, Afonso S. Bandeira, and Georgina Hall. 2016. “Exact Recovery in the Stochastic Block Model.” *IEEE Transactions on Information Theory / Professional Technical Group on Information Theory* 62 (1): 471–87.

- Abbe, Emmanuel, and Colin Sandon. 2015. "Recovering Communities in the General Stochastic Block Model without Knowing the Parameters." *arXiv [math.PR]*. arXiv. <http://arxiv.org/abs/1506.03729>.
- Aicher, Christopher, Abigail Z. Jacobs, and Aaron Clauset. 2013. "Adapting the Stochastic Block Model to Edge-Weighted Networks." *arXiv [stat.ML]*. arXiv. <http://arxiv.org/abs/1305.5782>.
- Altschul, S. F., T. L. Madden, A. A. Schäffer, J. Zhang, Z. Zhang, W. Miller, and D. J. Lipman. 1997. "Gapped BLAST and PSI-BLAST: A New Generation of Protein Database Search Programs." *Nucleic Acids Research* 25 (17): 3389–3402.
- Bornmann, Lutz, and Hans-dieter Daniel. 2008. "What Do Citation Counts Measure? A Review of Studies on Citing Behavior." *Journal of Documentation* 64 (1): 45–80.
- Brysbaert, Marc, and Sinead Smyth. 2011. "Self-Enhancement in Scientific Research: The Self-Citation Bias." *Psychologica Belgica* 51 (2): 129–37.
- Burt, Ronald S. 2004. "Structural Holes and Good Ideas." *The American Journal of Sociology* 110 (2): 349–99.
- . 2021. *Structural Holes*. Harvard University Press.
- Callon, Michel, Arie Rip, and John Law. 1986. *Mapping the Dynamics of Science and Technology: Sociology of Science in the Real World*. Springer.
- Case, Donald O., and Georgeann M. Higgins. 2000. "How Can We Investigate Citation Behavior? A Study of Reasons for Citing Literature in Communication." *Journal of the American Society for Information Science. American Society for Information Science* 51 (7): 635–45.
- Celisse, Alain, Jean-Jacques Daudin, and Laurent Pierre. 2012. "Consistency of Maximum-Likelihood and Variational Estimators in the Stochastic Block Model." *European Journal of Sport Science: EJSS: Official Journal of the European College of Sport Science* 6 (none): 1847–99.
- Decelle, Aurelien, Florent Krzakala, Cristopher Moore, and Lenka Zdeborová. 2011. "Asymptotic Analysis of the Stochastic Block Model for Modular Networks and Its Algorithmic Applications." *Physical Review. E, Statistical, Nonlinear, and Soft Matter Physics* 84 (6 Pt 2): 066106.
- Duede, Eamon, and James Evans. 2021. "The Social Abduction of Science." *arXiv [physics.soc-ph]*. arXiv. <http://arxiv.org/abs/2111.13251>.
- Ettensperger, Felix. 2020. "Comparing Supervised Learning Algorithms and Artificial Neural Networks for Conflict Prediction: Performance and Applicability of Deep Learning in the Field." *Quality & Quantity* 54 (2): 567–601.
- Etzkowitz, Henry, and Loet Leydesdorff. 2000. "The Dynamics of Innovation: From National Systems and 'Mode 2' to a Triple Helix of University–industry–government Relations." *Research Policy* 29 (2): 109–23.
- Fortunato, Santo, Carl T. Bergstrom, Katy Börner, James A. Evans, Dirk Helbing, Staša Milojević, Alexander M. Petersen, et al. 2018. "Science of Science." *Science* 359 (6379). <https://doi.org/10.1126/science.aao0185>.
- Gryniewicz, G., M. Poenie, and R. Y. Tsien. 1985. "A New Generation of Ca²⁺ Indicators with Greatly Improved Fluorescence Properties." *The Journal of Biological Chemistry* 260 (6): 3440–50.
- Guimerà, R., A. Díaz-Guilera, F. Vega-Redondo, A. Cabrales, and A. Arenas. 2002. "Optimal Network Topologies for Local Search with Congestion." *Physical Review*

Letters 89 (24): 248701.

- Jannot, Anne-Sophie, Thomas Agoritsas, Angèle Gayet-Ageron, and Thomas V. Perneger. 2013. "Citation Bias Favoring Statistically Significant Studies Was Present in Medical Research." *Journal of Clinical Epidemiology* 66 (3): 296–301.
- Jia, Tao, Dashun Wang, and Boleslaw K. Szymanski. 2017. "Quantifying Patterns of Research-Interest Evolution." *Nature Human Behaviour* 1 (4): 1–7.
- Kleinberg, Jon, Jens Ludwig, Sendhil Mullainathan, and Ziad Obermeyer. 2015. "Prediction Policy Problems." *The American Economic Review* 105 (5): 491–95.
- Latouche, Pierre, Etienne Birmelé, and Christophe Ambroise. 2011. "OVERLAPPING STOCHASTIC BLOCK MODELS WITH APPLICATION TO THE FRENCH POLITICAL BLOGOSPHERE." *The Annals of Applied Statistics* 5 (1): 309–36.
- Leahey, Erin, Christine M. Beckman, and Taryn L. Stanko. 2017. "Prominent but Less Productive: The Impact of Interdisciplinarity on Scientists' Research." *Administrative Science Quarterly* 62 (1): 105–39.
- Lehmann, Janette, Bruno Gonçalves, José J. Ramasco, and Ciro Cattuto. 2012. "Dynamical Classes of Collective Attention in Twitter." In *Proceedings of the 21st International Conference on World Wide Web*, 251–60. WWW '12. New York, NY, USA: Association for Computing Machinery.
- Lei, Jing, and Alessandro Rinaldo. 2015. "Consistency of Spectral Clustering in Stochastic Block Models." *The Annals of Statistics* 43 (1): 215–37.
- Leydesdorff, Loet. 2000. "The Triple Helix: An Evolutionary Model of Innovations." *Research Policy* 29 (2): 243–55.
- . 2007. "Betweenness Centrality as an Indicator of the Interdisciplinarity of Scientific Journals." *Journal of the American Society for Information Science and Technology* 58 (9): 1303–19.
- . 2010. "The Communication of Meaning and the Structuration of Expectations: Giddens' 'structuration Theory' and Luhmann's 'self-Organization.'" *Journal of the American Society for Information Science and Technology* 61 (10): 2138–50.
- Leydesdorff, Loet, and Ismael Rafols. 2011. "Indicators of the Interdisciplinarity of Journals: Diversity, Centrality, and Citations." *Journal of Informetrics* 5 (1): 87–100.
- Leydesdorff, L., and P. Van Den Besselaar. 1997. "Scientometrics and Communication Theory: Towards Theoretically Informed Indicators." *Scientometrics* 38 (1): 155–74.
- Li, Danielle, and Leila Agha. 2015. "Research Funding. Big Names or Big Ideas: Do Peer-Review Panels Select the Best Science Proposals?" *Science* 348 (6233): 434–38.
- Liu, Lu, Nima Dehmamy, Jillian Chown, C. Lee Giles, and Dashun Wang. 2021. "Understanding the Onset of Hot Streaks across Artistic, Cultural, and Scientific Careers." *Nature Communications*. <https://doi.org/10.1038/s41467-021-25477-8>.
- Liu, Lu, Yang Wang, Roberta Sinatra, C. Lee Giles, Chaoming Song, and Dashun Wang. 2018. "Hot Streaks in Artistic, Cultural, and Scientific Careers." *Nature* 559 (7714): 396–99.
- Matias, Catherine, and Vincent Miele. 2015. "Statistical Clustering of Temporal Networks through a Dynamic Stochastic Block Model." *arXiv [stat.ME]*. arXiv. <http://arxiv.org/abs/1506.07464>.
- Ma, Yifang, and Brian Uzzi. 2018. "Scientific Prize Network Predicts Who Pushes the Boundaries of Science." *Proceedings of the National Academy of Sciences of the*

- United States of America* 115 (50): 12608–15.
- Merton, Robert K. 1942. "The Normative Structure of Science In: RK Merton." *The Sociol.*
- . 1957. "Priorities in Scientific Discovery: A Chapter in the Sociology of Science." *American Sociological Review* 22 (6): 635–59.
- Mikolov, Tomas, Kai Chen, Greg Corrado, and Jeffrey Dean. 2013. "Efficient Estimation of Word Representations in Vector Space." *arXiv [cs.CL]*. arXiv. <http://arxiv.org/abs/1301.3781>.
- Mocanu, Delia, Luca Rossi, Qian Zhang, Marton Karsai, and Walter Quattrociocchi. 2015. "Collective Attention in the Age of (mis)information." *Computers in Human Behavior* 51 (October): 1198–1204.
- Molina, Mario, and Filiz Garip. 2019. "Machine Learning for Sociology." *Annual Review of Sociology*, July. <https://doi.org/10.1146/annurev-soc-073117-041106>.
- Morillo, Fernanda, Maria Bordons, and Isabel Gomez. 2001. *Scientometrics* 51 (1): 203–22.
- Morillo, Fernanda, María Bordons, and Isabel Gómez. 2003. "Interdisciplinarity in Science: A Tentative Typology of Disciplines and Research Areas." *Journal of the American Society for Information Science and Technology* 54 (13): 1237–49.
- Mossel, Elchanan, Joe Neeman, and Allan Sly. 2012. "Stochastic Block Models and Reconstruction." *arXiv [math.PR]*. arXiv. <http://arxiv.org/abs/1202.1499>.
- Nieminen, Pentti, Gerta Rucker, Jouko Miettunen, James Carpenter, and Martin Schumacher. 2007. "Statistically Significant Papers in Psychiatry Were Cited More Often than Others." *Journal of Clinical Epidemiology* 60 (9): 939–46.
- Partha, Dasgupta, and Paul A. David. 1994. "Toward a New Economics of Science." *Research Policy* 23 (5): 487–521.
- Porter, Alan L., Alex S. Cohen, J. David Roessner, and Marty Perreault. 2007. "Measuring Researcher Interdisciplinarity." *Scientometrics* 72 (1): 117–47.
- Porter, Alan L., and Ismael Rafols. 2009. "Is Science Becoming More Interdisciplinary? Measuring and Mapping Six Research Fields over Time." *Scientometrics* 81 (3): 719–45.
- Rachel Wang, Y. X., and Peter J. Bickel. 2017. "Likelihood-Based Model Selection for Stochastic Block Models." *The Annals of Statistics* 45 (2): 500–528.
- Rafols, Ismael, Loet Leydesdorff, Alice O'Hare, Paul Nightingale, and Andy Stirling. 2012. "How Journal Rankings Can Suppress Interdisciplinary Research: A Comparison between Innovation Studies and Business & Management." *Research Policy* 41 (7): 1262–82.
- Rheingold, Howard. 1985. *Tools for Thought: The People and Ideas behind the Next Computer Revolution*. Simon & Schuster Trade.
- Rigby, John. 2013. "Looking for the Impact of Peer Review: Does Count of Funding Acknowledgements Really Predict Research Impact?" *Scientometrics* 94 (1): 57–73.
- Selin, Henrik, and Stacy D. VanDeveer. 2007. "Political Science and Prediction: What's next for U.s. Climate Change Policy?" *The Review of Policy Research* 24 (1): 1–27.
- Simandan, Dragos. 2020. "Being Surprised and Surprising Ourselves: A Geography of Personal and Social Change." *Progress in Human Geography* 44 (1): 99–118.
- Sinatra, Roberta, Dashun Wang, Pierre Deville, Chaoming Song, and Albert-László

- Barabási. 2016. "Quantifying the Evolution of Individual Scientific Impact." *Science* 354 (6312). <https://doi.org/10.1126/science.aaf5239>.
- Stanley, Natalie, Saray Shai, Dane Taylor, and Peter J. Mucha. 2016. "Clustering Network Layers with the Strata Multilayer Stochastic Block Model." *IEEE Transactions on Network Science and Engineering* 3 (2): 95–105.
- Stuart, Toby E., and Joel M. Podolny. 2007. "Local Search and the Evolution of Technological Capabilities." *Strategic Management Journal* 17 (S1): 21–38.
- Szell, Michael, Yifang Ma, and Roberta Sinatra. 2018. "A Nobel Opportunity for Interdisciplinarity." *Nature Physics* 14 (11): 1075–78.
- Tshitoyan, Vahe, John Dagdelen, Leigh Weston, Alexander Dunn, Ziqin Rong, Olga Kononova, Kristin A. Persson, Gerbrand Ceder, and Anubhav Jain. 2019. "Unsupervised Word Embeddings Capture Latent Knowledge from Materials Science Literature." *Nature* 571 (7763): 95–98.
- Uzzi, Brian, Satyam Mukherjee, Michael Stringer, and Ben Jones. 2013. "Atypical Combinations and Scientific Impact." *Science* 342 (6157): 468–72.
- Uzzi, Brian, and Jarrett Spiro. 2005. "Collaboration and Creativity: The Small World Problem." *The American Journal of Sociology* 111 (2): 447–504.
- Vaan, Mathijs de, Balazs Vedres, and David Stark. 2015. "Game Changer: The Topology of Creativity1." *AJS; American Journal of Sociology* 120 (4): 1–51.
- Van Noorden, Richard, Brendan Maher, and Regina Nuzzo. 2014. "The Top 100 Papers." *Nature* 514 (7524): 550–53.
- Wagner, Caroline S., Edwin Hurlings, Travis A. Whetsell, Pauline Mattsson, and Katarina Nordqvist. 2015. "Correction: Do Nobel Laureates Create Prize-Winning Networks? An Analysis of Collaborative Research in Physiology or Medicine." *PloS One* 10 (8): e0136478.
- Wang, Dashun, Chaoming Song, and Albert-László Barabási. 2013. "Quantifying Long-Term Scientific Impact." *Science* 342 (6154): 127–32.
- Wang, Jian, Bart Thijs, and Wolfgang Glänzel. 2015. "Interdisciplinarity and Impact: Distinct Effects of Variety, Balance, and Disparity." *PloS One* 10 (5): e0127298.
- Wang, Qi, and Jesper Wiborg Schneider. 2020. "Consistency and Validity of Interdisciplinarity Measures." *Quantitative Science Studies* 1 (1): 239–63.
- Wu, Fang, and Bernardo A. Huberman. 2007. "Novelty and Collective Attention." *Proceedings of the National Academy of Sciences of the United States of America* 104 (45): 17599–601.
- Yanagisawa, M., H. Kurihara, S. Kimura, Y. Tomobe, M. Kobayashi, Y. Mitsui, Y. Yazaki, K. Goto, and T. Masaki. 1988. "A Novel Potent Vasoconstrictor Peptide Produced by Vascular Endothelial Cells." *Nature* 332 (6163): 411–15.
- Zeng, An, Zhesi Shen, Jianlin Zhou, Ying Fan, Zengru Di, Yougui Wang, H. Eugene Stanley, and Shlomo Havlin. 2019. "Increasing Trend of Scientists to Switch between Topics." *Nature Communications* 10 (1): 3439.
- Zeng, Langche. 1999. "Prediction and Classification with Neural Network Models." *Sociological Methods & Research* 27 (4): 499–524.
- Zhang, Lin, Ronald Rousseau, and Wolfgang Glänzel. 2016. "Diversity of References as an Indicator of the Interdisciplinarity of Journals: Taking Similarity between Subject Fields into Account." *Journal of the Association for Information Science and Technology* 67 (5): 1257–65.

Zuckerman, Harriet. 1978. "Views: The Sociology of the Nobel Prize: Further Notes and Queries: How Successful Are the Prizes in Recognizing Scientific Excellence?" *American Scientist* 66 (4): 420–25.

REVIEWERS' COMMENTS

Reviewer #1 (Remarks to the Author):

Thank you for your careful response. The revision has addressed my concerns and thus I recommend the publication of the manuscript.

Reviewer #2 (Remarks to the Author):

Thank you. I am satisfied that the author(s) have responded thoroughly to the comments made by the reviewer. Excellent job.

Reviewer #3 (Remarks to the Author):

The authors have sufficiently addressed my primary concerns, including surprise examples to anchor the model, methodological clarifications, overstatements of the results, and normalization issues. I am still concerned about the strength of the claims without the papers that never make it to publication, but that can wait for another analysis, given the difficulty of obtaining this kind of data.